# App-based COVID-19 syndromic surveillance and prediction of hospital admissions in COVID Symptom Study Sweden

Beatrice Kennedy[1], Hugo Fitipaldi [2], Ulf Hammar[1], Marlena Maziarz [3], Neli Tsereteli[2], Nikolay Oskolkov[4], Georgios Varotsis[1], Camilla A. Franks[3], Diem Nguyen [1], Lampros Spiliopoulos[3,5], Hans-Olov Adami [6,7,8], Jonas Björk [9,10], Stefan Engblom [11], Katja Fall[12,13], Anna Grimby-Ekman[14], Jan-Eric Litton[7], Mats Martinell[15,16], Anna Oudin[9,17], Torbjörn Sjöström[18], Toomas Timpka [19], Carole H. Sudre[20,21,22], Mark S. Graham[22], Julien Lavigne du Cadet[23], Andrew T. Chan[24], Richard Davies [23], Sajaysurya Ganesh[23], Anna May[23], Sébastien Ourselin[22], Joan Capdevila Pujol[23], Somesh Selvachandran[23], Jonathan Wolf [23], Tim D. Spector [25], Claire J. Steves [25], Maria F. Gomez [3,26], Paul W. Franks [2,26] & Tove Fall [1,26✉]

The app-based COVID Symptom Study was launched in Sweden in April 2020 to contribute to real-time COVID-19 surveillance. We enrolled 143,531 study participants (≥18 years) who contributed 10.6 million daily symptom reports between April 29, 2020 and February 10, 2021. Here, we include data from 19,161 self-reported PCR tests to create a symptom-based model to estimate the individual probability of symptomatic COVID-19, with an AUC of 0.78 (95% CI 0.74–0.83) in an external dataset. These individual probabilities are employed to estimate daily regional COVID-19 prevalence, which are in turn used together with current hospital data to predict next week COVID-19 hospital admissions. We show that this hospital prediction model demonstrates a lower median absolute percentage error (MdAPE: 25.9%) across the five most populated regions in Sweden during the first pandemic wave than a model based on case notifications (MdAPE: 30.3%). During the second wave, the error rates are similar. When we apply the same model to an English dataset, not including local COVID-19 test data, we observe MdAPEs of 22.3% and 19.0% during the first and second pandemic waves, respectively, highlighting the transferability of the prediction model.

A full list of author affiliations appears at the end of the paper.

Real-time and accurate COVID-19 disease surveillance data is critical for adequate public health decision making and evaluation, as well as for healthcare system preparedness. The WHO guidelines for COVID-19 surveillance highlight the importance of combining data from multiple surveillance systems, and how participatory syndromic surveillance, where participants self-report symptoms of possible infection, may constitute a useful tool in early detection of disease outbreaks[1]. The European Centre for Disease Prevention and Control further notes that the utility of COVID-19 participatory syndromic surveillance may be enhanced if symptom data can be combined with information on testing[2]. Expanding knowledge on the feasibility of large-scale syndromic surveillance may thus enable tailored population-based participatory surveillance initiatives in future pandemics and epidemics.

Several novel eHealth solutions aimed at real-time monitoring and prediction of the dynamics of COVID-19 transmission were introduced early in the pandemic[3–6], with app-based technologies quickly recognized as a potentially powerful surveillance tool. One of these technologies was the ZOE COVID Symptom Study app, designed to collect baseline health data as well as daily reports on symptoms and test results from study participants. The app was launched in the United Kingdom and in the United States in late March 2020[7–9].

Community transmission of SARS-CoV-2 was confirmed in Sweden in early March 2020. However, during the first pandemic wave in the spring of 2020, PCR testing was only available for hospital inpatients and healthcare workers[10] in Sweden and assessments of national COVID-19 prevalence were based on two PCR surveys performed by the Public Health Agency of Sweden in April (n = 2571) and May (n = 2957)[11]. Nationwide PCR testing for symptomatic adults was later introduced in June 2020[10], but suffered from various issues such as long analysis times during periods of high demand[12]. In response to the limited surveillance during the first pandemic wave, the COVID Symptom Study was launched in Sweden in April 2020.

The aims of this study were to develop and evaluate a syndromic surveillance-based framework to estimate the regional prevalence of COVID-19 and to evaluate if these could be used to accurately predict subsequent trends in COVID-19 hospital admissions. We showed that a model trained on symptoms and test data could provide informative prevalence estimates, and contribute to predictions of hospital burden the following week. Without using any additional test data, the hospital prediction model further performed well outside Sweden in a second country, England.

## Results

In this nationwide study on COVID-19 during the first year of the pandemic, we included data from 143,531 COVID Symptom Study Sweden (CSSS) participants ≥18 years from April 29, 2020 to February 10, 2021, who had contributed with at least one daily report in the app (Supplementary Table 1). The median duration of study participation was 151 days (inter-quartile range [IQR] 52–252), with a median of 43 days with submitted reports (IQR 13–119). Of all participants, 30% reported at least one COVID-19 PCR test, and 6% of women and 4% of men reported at least one positive test result. The cohort included a larger proportion of women, fewer people ≥65 years, and fewer smokers than the general population, while the prevalence of obesity was similar. Participants further resided in postal code areas with less deprivation, similar proportions of inhabitants with foreign background, and higher population densities, as compared to the general population. The frequency of participants across regions is depicted in Supplementary Fig. 1. CSSS was led by researchers at Lund University and Uppsala University, and the highest CSSS participation rates were observed in the regions of Skåne and Uppsala where these universities are located, as well in the most populated region in Sweden, Stockholm (Supplementary Table 2).

The most common symptoms reported in participants with PCR-confirmed COVID-19 were loss of smell and/or taste, headache, fever, and sore throat (Fig. 1a). The prevalence of loss of smell and/or taste peaked at 4 days after the test date. Among participants who tested negative, headache and sore throat were most common, whereas loss of smell and/or taste was rarely reported (Fig. 1b). The non-adjusted prevalence of different symptoms was considerably higher in the symptom data collected

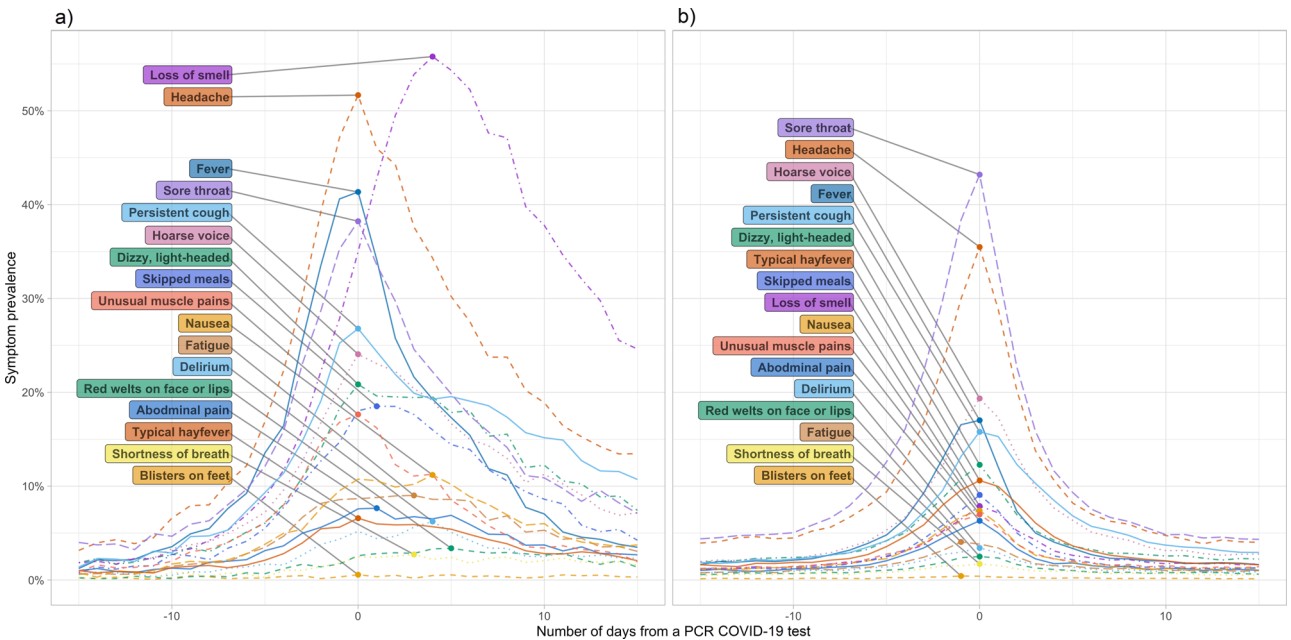

**Fig. 1 Symptom trajectories.** The prevalence of symptoms reported by participants in COVID Symptom Study Sweden with (**a**) a positive PCR test for COVID-19 (n = 5178), and (**b**) a negative PCR test for COVID-19 (n = 32,089), across the study period April 29, 2020–February 10, 2021.

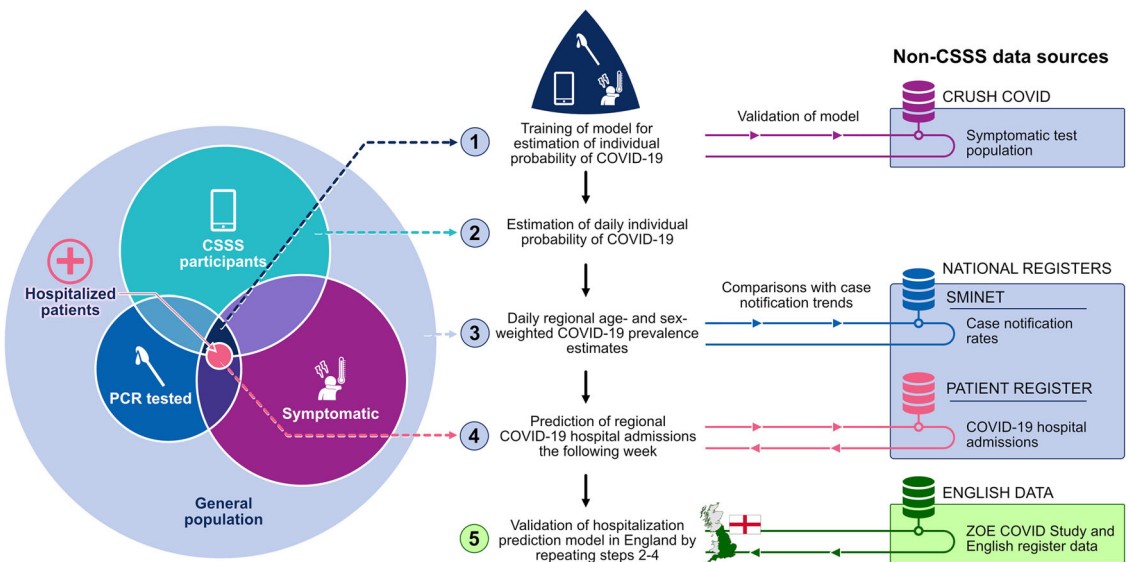

**Fig. 2 Analysis strategy.** Analysis strategy and data sources.

by the Swedish company NOVUS than in CSSS, with the exception of loss of smell and/or taste, but temporal trends were similar (Supplementary Fig. 2). This discrepancy was likely due to the different approach in CSSS, where participants were asked questions about symptoms only if they responded negative to the first gate-keeper question, namely whether they felt healthy as normal.

Step 1. Training of model for estimation of individual probability of COVID-19

Our analysis strategy consisted of five steps, as illustrated in Fig. 2. In the first step, we developed a model to estimate individual probability of symptomatic COVID-19, utilizing information from 19,161 CSSS participants who reported at least one PCR test (of whom 2586 were COVID-19 infection positive) between April 29 and December 31, 2020; these individuals also reported at least one candidate symptom within 7 days before or on the test date. The final model selected by LASSO included 17 symptoms and sex, as well as two-way interactions between loss of smell and/or taste and 14 symptoms, and a two-way interaction between loss of smell and/or taste and sex. The ROC area under the curve (AUC) for this main model was 0.76 (95% CI 0.75–0.78; PR(AUC) with 95% CI: 0.38, 95% CI 0.35–0.40) during the training period (April 29–December 31, 2020; $n = 19,161$) and 0.72 (95% CI 0.69–0.75; PR(AUC) 0.40, 95% CI 0.35–0.45) during the evaluation period (January 1–February 10, 2021; $n = 1753$). The AUC for the training period was produced by nested tenfold cross-validation. In an external dataset of 943 symptomatic individuals from the CRUSH Covid survey (144 positive; test positivity 15.3%; October 18, 2020–February 10, 2021), the AUC was 0.78 (95% CI 0.74–0.83; PR(AUC): 0.48, 95% CI 0.40–0.56). Calibration graphs are available in Supplementary Fig. 3.

Step 2. Estimation of daily individual probability of COVID-19 in CSSS

We applied the model from Step 1 to estimate the daily individual probability of symptomatic COVID-19 in all CSSS study participants, including non-tested individuals, across the entire study period from May 10, 2020, through February 10, 2021. Non-symptomatic individuals were assigned a probability of 0 for that day.

Step 3. Daily regional COVID-19 prevalence estimates in the general population

The individual probabilities from Step 2 were then used to estimate the daily regional COVID-19 prevalence in the general population in Sweden, accounting for differences in age and sex distributions of the participants as compared to the general population in each region. We calibrated the intercept of the model generated in step 1 so that the estimated prevalence of May 27, 2020 matched the estimated prevalence from a national COVID-19 prevalence survey ($n = 2957$). The resulting CSSS prevalence estimates of symptomatic COVID-19 showed similar waves as the first and second waves of COVID-19 hospitalization (Fig. 3a, Supplementary Fig. 4a). In contrast, data from SmiNet, the national Swedish register on laboratory-confirmed cases of COVID-19, did not capture the first wave (Fig. 3b, Supplementary Fig. 4b).

During the autumn of 2020, we observed a peak in CSSS-based COVID-19 prevalence estimates in September 2020 with no corresponding peak in other COVID-19 national case notification rates or hospital admission data. We therefore constructed a retrospective time-dependent model for individual probability of symptomatic COVID-19, based on the main model in Step 1 and validated correspondingly (additional information and calibration graphs available in the Supplementary Material). Retrospective national COVID-19 prevalence estimates based on the time-dependent model showed higher concordance with national COVID-19 case notification and hospital admission trends than the main model (Fig. 3c, d).

We further observed a higher estimated prevalence of symptomatic COVID-19 in women than in men across the entire study period, which was most apparent in those ≤64 years (Supplementary Fig. 5a). Post-hoc analyses revealed that this difference was mainly generated by participants who were healthcare professionals, where women were over-represented (Supplementary Fig. 5b).

Step 4. Prediction of regional COVID-19 hospital admissions the following week

We developed an iterative time-updated prediction model to assess if the regional prevalence estimates from Step 3 could be used together with current hospital data for prediction of regional COVID-19 hospital admissions 7 days ahead. The parameters were estimated on available data through June 1, 2020 with larger weights applied to more recent observations, to predict admissions on June 8, 2020. This procedure was repeated to calibrate

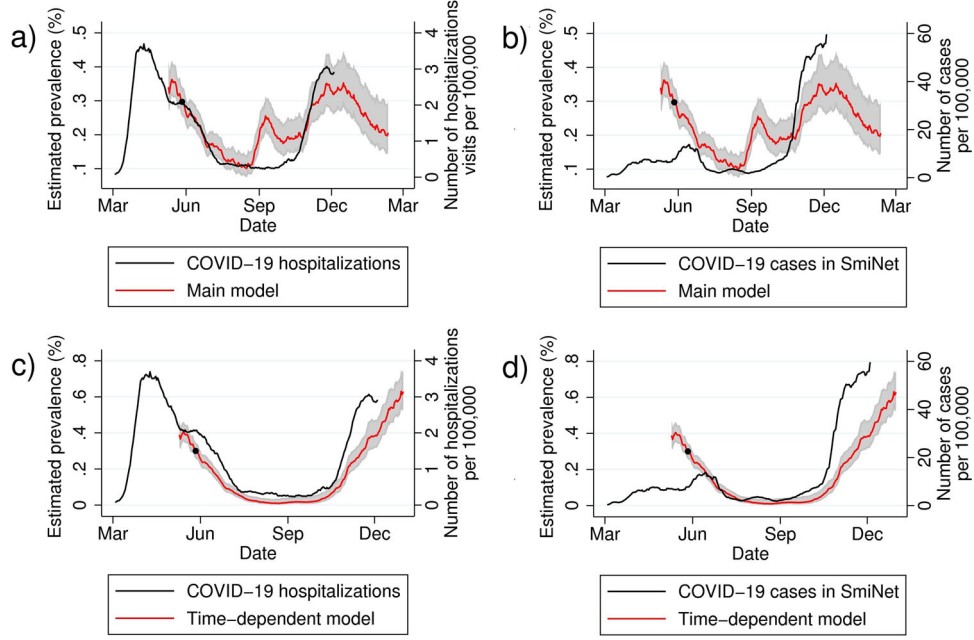

**Fig. 3 Prevalence estimates of symptomatic COVID-19 in Sweden.** National prevalence estimate, with 95% confidence interval, of symptomatic COVID-19 in COVID Symptom Study Sweden (main model utilized for real-time prediction estimates, and retrospective time-dependent model), combined in (**a** and **c**) with retrospective data on daily number of new hospital admissions registered in the National Patient Register per 100,000 inhabitants ≥18 years, and in (**b** and **d**) with daily number of new COVID-19 cases registered in SmiNet, per 100,000 inhabitants ≥18 years. *Time-point for recalibration of CSSS national COVID-19 prevalence estimate using national point prevalence survey findings from the Public Health Agency of Sweden.

the coefficients throughout the study period. Weights and model specifications were based on exploratory analyses using data from May 11 to November 29, 2020. Overall, 16,752 individuals (≥18 years) were admitted to hospital with a diagnosis of COVID-19 from May 11 through November 29, 2020, and the number of daily new COVID-19 hospital admissions ranged from 0 to 104 across the 21 Swedish healthcare regions.

Across the five most populated regions in Sweden, the CSSS hospital prediction model demonstrated a median absolute percentage error (MdAPE) of 25.9% between daily predicted and observed number of hospital admissions for the first pandemic wave (June 8–July 3, 2020; Fig. 4 and Table 1), while the MdAPE for the second wave (October 19–November 29, 2020) was 26.8%, which was lower than, or similar to, the predictions from a similar prediction model combining daily case notifications from SmiNet with hospital admissions yielded MdAPEs of 30.3% and 25.9% for the five most populated regions (first and second wave, respectively; Supplementary Table 3).

The MdAPEs were lowest in the most populated region in Sweden (Stockholm, population ≥18 years $n = 1.85$ million) with 12.2% and 16.6% (SmiNet-based MdAPEs 13.5% and 24.0%) for the first and second waves, respectively. When we pooled data from all 21 Swedish regions, MdAPEs for the first and second waves were higher for both the CSSS hospital prediction model (37.0% and 42.4%, respectively; Supplementary Fig. 6) and the SmiNet-based model (38.7% and 38.5%, respectively). Overall, we noted that the accuracy of the hospital prediction model as measured on the relative scale was lower when regional daily number of hospital admissions was low (Supplementary Fig. 7a).

Step 5. Validation of the hospitalization prediction model in England

We sought to validate the CSSS-based hospitalization prediction model in England by repeating Steps 2 and 3 and parts of Step 4. The English dataset encompassed daily reports from 2,638,536 ZOE COVID Study English study participants from March 30, 2020 to January 31, 2021 (study population

characteristics and regional participation rates are available in Supplementary Tables 4 and 5). We extracted information on all COVID-19 hospital admissions ($n = 318,232$) in individuals ≥18 years across the seven English healthcare regions from April 6, 2020 to February 7, 2021 from National Health Service England data. The number of new daily COVID-19 hospital admissions ranged from 0 to 958 across the English regions during this period.

We applied the exact same model that was developed in Step 1 in CSSS to estimate daily individual probability of COVID-19 in the English dataset, and then estimated the daily age- and sex-weighted COVID-19 prevalence across the English regions. We further applied the same iterative time-updated prediction model as in the Swedish dataset to predict hospital admissions the following week in the seven English regions. We used available outcome data up to April 27, 2020 to tune the parameters and to predict admissions a week later on May 4, 2020 and then repeated this daily throughout the study period (until February 7, 2021).

Across the seven English regions, we observed an MdAPE of 22.3% for the part of the first English pandemic wave captured in the data (May 4–June 19, 2020) and an MdAPE of 19.0% for the second English wave (September 20, 2020–February 7, 2021; Fig. 5, Supplementary Table 6). As in Sweden, we observe lower error in the most populated English healthcare region (West and East Midlands; population ≥18 years $n = 10.8$ million) with MdAPE of 16.1% and 14.0%, respectively. Overall, the predicted number of hospital admissions were overestimated when daily regional hospital admissions were low (Supplementary Fig. 7b).

## Discussion

Adequate and continuous regional COVID-19 surveillance is challenging and requires multiple sources of data. Here, we developed an app-based framework that allowed for syndromic surveillance of COVID-19 at national and regional level in Sweden across the first two pandemic waves. We found that CSSS

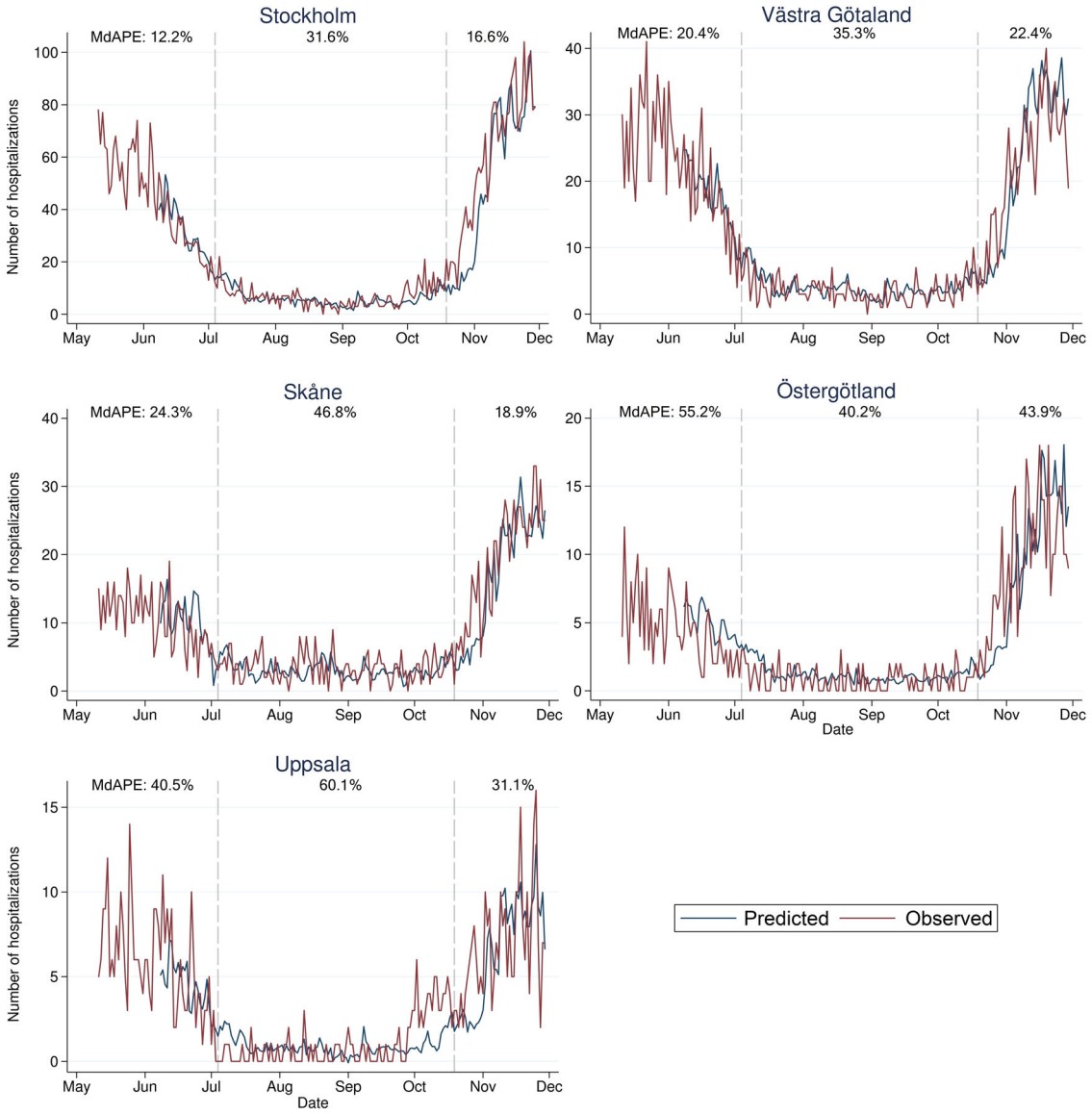

**Fig. 4 Predicted number of daily hospital admissions in Sweden.** Predicted number of daily hospital admissions 7 days ahead across the five most populated regions in Sweden ordered by population size. The median absolute percentage errors (MdAPEs) of the predictions are denoted for the first pandemic wave (June 8–July 3, 2020), the summer period (July 4–October 18, 2020), and the second pandemic wave (October 19–November 29, 2020).

prevalence estimates could be used to monitor COVID-19 disease trends, and that they were particularly informative in times of limited PCR testing capacity. The accuracy of the prevalence estimates was, however, lower in September 2020 when other respiratory infections peaked. We further showed that combining app-based regional prevalence estimates with previously recorded hospital data could, with moderate accuracy, predict regional levels of COVID-19 hospital admissions the following week both in Sweden and in England.

A previous study from ZOE COVID Study demonstrated how app data from the first pandemic wave from March through September 2020 could be utilized to successfully identify emerging COVID-19 hotpots in England, with findings validated in UK Government test data[13]. The present study confirmed the utility of app-based COVID-19 syndromic surveillance, encompassing the full second pandemic wave in the Swedish population and expanding the scope to include predictions of subsequent hospital admissions. The validation of the CSSS-based hospital prediction model in English data highlights the potential transferability of our approach, without using any PCR test data in the

English data. Syndromic surveillance of COVID-19 may thus provide early warnings of surges in hospital admissions, thereby helping guide the allocation of precious healthcare resources in times of crisis.

A strength of the CSSS was that the syndromic surveillance using daily reports from study participants, which enables rapid data analysis and quick dissemination of results. The prevalence estimates were continuously disseminated to the study participants and the general public via the CSSS dashboard[14]. In contrast, the official COVID-19 disease surveillance in Sweden has suffered delays even after PCR testing was made available to the general public in June 2020. The time interval from booking a PCR test to confirmation of test result exceeded 6 days across several regions during our study period[12], with additional time delays in the reporting of COVID-19 notification rates on regional and municipality levels by the Public Health Agency of Sweden[15].

The accuracy of the CSSS-based hospital prediction model during the first and second wave was higher for more populated regions in Sweden. Moreover, when the Swedish model was

**Table 1 Median absolute percentage errors (MdAPEs) for prediction of new daily hospitalizations, across the first pandemic wave (June 8–July 3, 2020), the summer period (July 4–October 18, 2020), and the second pandemic wave (October 19–November 29, 2020).**

| | Median absolute percentage errors (%) | | |
| --- | --- | --- | --- |
| | First wave June 8–July 3, 2020 | Summer period July 4–October 18, 2020 | Second wave October 19–November 29, 2020 |
| All 21 regions combined | 37.0 | 48.2 | 42.4 |
| Top 5 most populated regions[a] | 25.9 | 38.6 | 26.8 |
| Blekinge | 41.4 | 67.6 | 55.8 |
| Dalarna | 58.5 | 49.4 | 47.7 |
| Gotland | 48.3 | 83.5 | 49.2 |
| Gävleborg | 29.4 | 54.1 | 44.0 |
| Halland | 39.4 | 49.8 | 73.4 |
| Jämtland | 31.9 | 74.5 | 51.6 |
| Jönköping | 32.4 | 45.5 | 28.5 |
| Kalmar | 52.5 | 53.7 | 44.7 |
| Kronoberg | 53.4 | 69.4 | 52.5 |
| Norrbotten | 40.3 | 50.3 | 68.6 |
| Skåne | 24.3 | 46.8 | 18.9 |
| Stockholm | 12.2 | 31.6 | 16.6 |
| Södermanland | 54.2 | 37.6 | 44.8 |
| Uppsala | 40.5 | 60.1 | 31.1 |
| Värmland | 41.0 | 55.1 | 43.7 |
| Västerbotten | 60.9 | 48.2 | 34.2 |
| Västernorrland | 36.3 | 54.1 | 42.9 |
| Västmanland | 39.3 | 49.1 | 30.2 |
| Västra Götaland | 20.4 | 35.3 | 22.4 |
| Örebro | 27.0 | 46.1 | 67.9 |
| Östergötland | 55.2 | 40.2 | 43.9 |

The iterative prediction model used current regional COVID Symptom Study Sweden prevalence estimates and hospital data to predict hospital admissions 7 days ahead.
[a]Stockholm, Västra Götaland, Skåne, Östergötland, Uppsala.

applied in England across healthcare regions, MdAPEs were lower than those derived in the Swedish setting. Although we cannot discern the separate influences of larger total population size, higher absolute number of study participants, and higher study participation rates, it is likely that all these factors enhance the accuracy of the hospital prediction model.

When we compared the CSSS-based hospital prediction model with the SmiNet-based (PCR test-based) hospital prediction model, we observed that the accuracy of the CSSS-based model was higher during the first wave, while the SmiNet-based model was similar in the second wave. The higher accuracy of CSSS during the first pandemic wave is likely due to the limited availability of PCR testing in Sweden at that time, when tests where only available to hospital inpatients and healthcare workers.[10] We conclude that in addition to the expansion of the national PCR testing programme introduced in June 2020 and subsequent delays in PCR testing, local factors may also influence how well the CSSS app and PCR testing efforts reflect regional and/or local outbreaks. However, the successful application of the non-test dependent hospital prediction model in England shows great promise for future efforts in syndromic surveillance, where models can be trained in one country with dense test data and adjusted to local trends of hospitalizations in a second country without the need for additional test data.

More than 166,000 participants (2.4% of the adult population in Sweden) joined CSSS in the first 5 weeks after launch, supporting the feasibility of large-scale app-based syndromic surveillance and the power of citizen science, which can be rapidly scaled-up without needing additional staff or large financial resources. However, although the use of a smart device app was intended to minimize barriers to enrolment[16], a lesser proportion of CSSS participants were male, ≥65 years, smokers, or obese, as compared to the general population, indicating an over-representation of healthy individuals. Further, owing to limited resources the CSSS app was in Sweden only available in Swedish, which precluded participation of non-Swedish speakers, a group that may be at higher risk of COVID-19 infection[17,18]. We also observed that the CSSS participants resided in postal code areas with on average less deprivation than the general public. Together, these factors may have limited the ability of the CSSS to detect local outbreaks in vulnerable neighborhoods, areas where lower community testing rates were also observed during the pandemic[12]. In future epidemic surveillance efforts, combining syndromic surveillance with non-participatory data sources, such as number of calls regarding specific complaints to nurse telephone consultation services[19], measurements of virus occurrence in wastewater[20,21], monitoring of mobility patters in the population[22], and aggregate data on vaccination rates across neighborhoods, may constitute a cost-efficient way to characterize community infection trends and predict increased demands on healthcare resources.

Additional potential limitations to the data collection also apply. Firstly, participants may have been more likely to join the study and report daily if they experienced symptoms perceived to be linked with COVID-19 than if they had been healthy, potentially inflating COVID-19 prevalence estimates. We sought to reduce this risk by excluding the first 7 days of data collected for each participant, but we cannot exclude the risk of residual participation bias. Secondly, all data collected in the COVID Symptom Study app are self-reported. We had no means of linking the self-reported data to national population or health registers, so we could not validate information on COVID-19 tests or baseline health survey questions. We were able to partly overcome this limitation by validating the model predicting COVID-19 PCR test positivity in the independent CRUSH Covid dataset. Lastly, some questions in the COVID Symptom Study app were updated during the study period, potentially influencing data collection and analyses. The baseline health survey questions that were modified and/or updated during this time were, however, not included in our analyses and did not affect the prevalence estimates or the accuracy of the hospitalization prediction model. The symptom questions were updated on November 4, 2020. This update was implemented at the same time as incidence was increasing sharply, but the national prevalence estimate curve and the regional prevalence estimate curves remained smooth, indicating that these updates did not materially influence our findings.

We observed a peak in app-based COVID-19 prevalence estimates in mid-September 2020 with no corresponding peak in any disease-specific COVID-19 national register data. The prevalence of loss of smell and/or taste, sore throat, and headache were similarly elevated in NOVUS. The Public Health Agency of Sweden also noted high occurrence of symptoms of acute respiratory infections at this time[23]. Laboratory analyses of respiratory viruses later indicated a high incidence of common colds caused by rhinoviruses in September 2020[24]. Hence, the specificity of the CSSS data was compromised when prevalence of other pathogens with similar symptomatology to COVID-19 was elevated. We therefore added time as a variable in the model developed in Step 1, allowing estimated probabilities to vary depending on the PCR test positivity rate during a given period. This second model yielded results more consistent with the

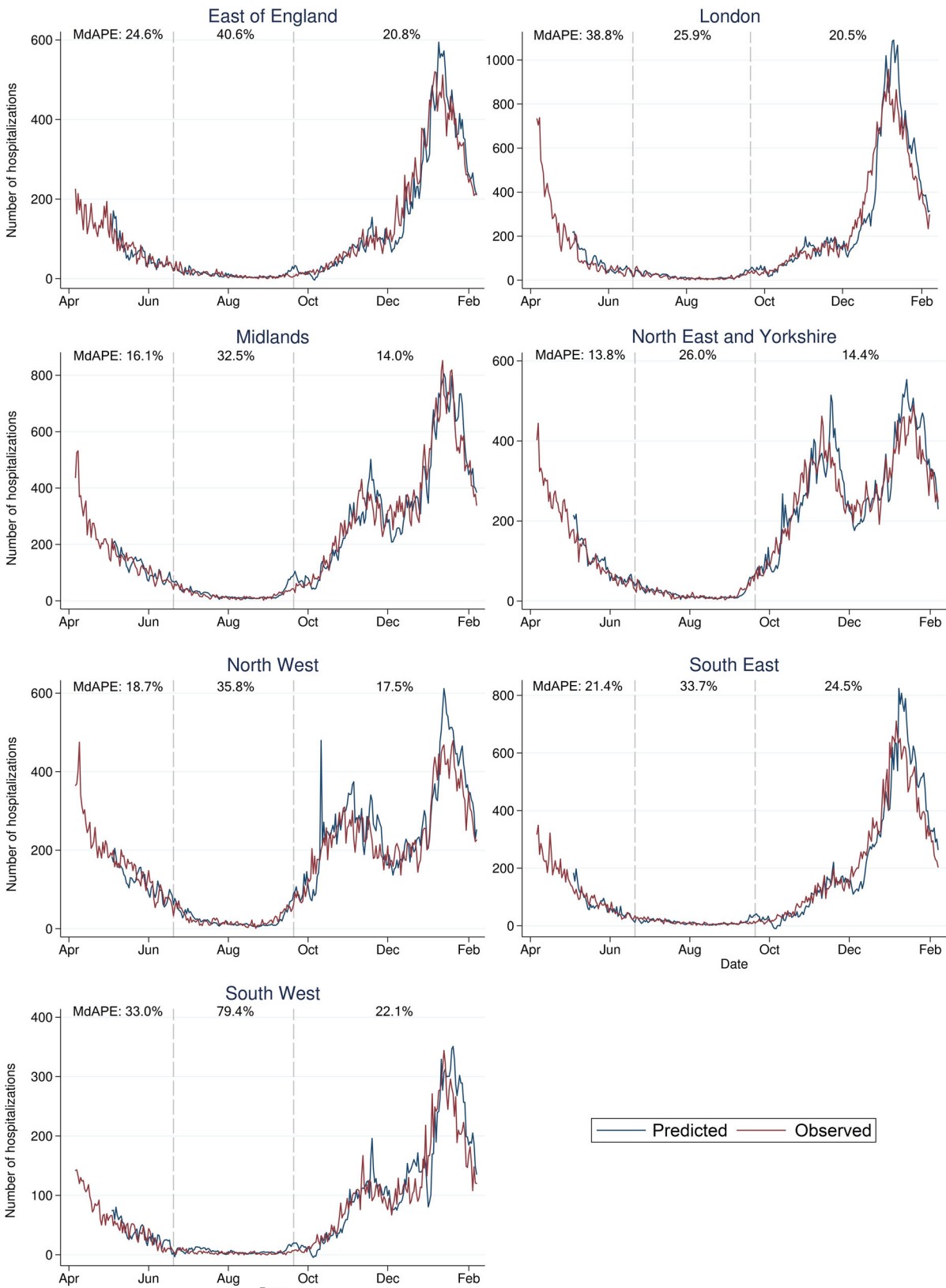

**Fig. 5 Predicted number of daily hospital admissions in England.** Predicted number of daily hospital admissions 7 days ahead across the seven English healthcare regions. The median absolute percentage errors (MdAPEs) of the predictions are denoted for the first pandemic wave (May 4–June 19, 2020), the summer period (June 20–September 19, 2020), and the second pandemic wave (September 20, 2020–February 7, 2021).

national COVID-19 incidence data but is strongly influenced by the proportion of positive COVID-19 tests. The model is thus rather insensitive to a sudden increase or decrease in the reporting of symptoms, and change in prevalence will not be fully captured until this trend is also reflected in the test results reported in the app. The more static main model will, conversely, react more quickly to an increase of reported symptoms, raising the sensitivity of the model but also the risk of false positive healthcare alerts. Because of the delay inherent in this type of analysis, the time-dependent model is not suitable for real-time COVID-19 surveillance; it is also ineffective when test positivity varies greatly across regions, unless regions are modelled separately. An intermediate solution would be to retrain the model at known events that affect the relationship between symptoms and positive PCR tests, such as when vaccinations are introduced, new variants or other concurrent epidemics emerge. Very few cases of seasonal influenza were confirmed in Sweden in the winter season of 2020/2021 compared with previous years[25], which rendered the lower specificity during this period less problematic.

As the study period ended in February 2021, this study encompassed pandemic waves characterized in Sweden mainly by the early SARS-CoV-2 strains, with the variant of concern Alpha being detected in late December 2020[26]. Moreover, <150,000 inhabitants (<2% of the total population) had been inoculated with two doses of vaccine by early February 2021[27]. Further studies on prediction of COVID-19 hospital admissions, including subsequent variants of concern as well as higher vaccination rates in the general population, are therefore warranted.

In conclusion, app-based syndromic surveillance and citizen science may represent a powerful and rapid asset when assessing the early spread of a pandemic virus. The findings from CSSS and validation in the English setting suggest that app-based technologies could contribute to national and regional disease surveillance and early warnings to healthcare systems.

## Methods

**Ethical approvals.** The Swedish Ethical Review Authority has approved CSSS (DNR 2020-01803 with addendums 2020-04006, 2020-04145, 2020-04451, 2020-07080, and 2021-02316) and CRUSH Covid (DNR 2020-07080, and DNR 2020-04210 with addendum 2020-06315). In the United Kingdom, the ZOE COVID Study has been approved by King's College London (KCL) ethics committee REMAS ID 18210, review reference LRS-19/20-18210. All participants in the CSSS and in ZOE COVID Study in the UK have provided informed consent. Participants have not been compensated for their participation.

**COVID symptom study sweden.** COVID Symptom Study Sweden (CSSS) was launched in Sweden on April 29, 2020 to provide COVID-19 syndromic surveillance data and to build a large-scale repeated measures database for COVID-19 research. More than 166,000 participants (2.4% of the adult population) joined CSSS in the first 5 weeks after launch. The non-commercial mobile application used in the study was initially developed by health data science company ZOE Limited in partnership with KCL and Massachusetts General Hospital[7,8], and adapted for use in Sweden by ZOE Limited in collaboration with Lund University and Uppsala University. The app has been used to study the contemporary disease burden and to predict consequences of COVID-19[9,13,28].

All individuals ≥18 years living in Sweden with access to a smart device were eligible to participate in the CSSS after downloading the app and providing informed consent. Participants are asked to report year of birth, sex, height, weight, postal code, whether they work in the healthcare sector, and to complete a health survey including pre-existing health conditions. Subsequently, participants were asked daily (with voluntary response frequency) if they felt "healthy as normal" or not, and to report the date and result of any COVID-19 PCR or serology test. If they did not feel healthy, they were asked about an array of symptoms potentially associated with COVID-19. The symptoms that participants could report included, but were not limited to, loss of smell and/or taste, fever, persistent cough, fatigue, abdominal pain, chest pain, hoarse voice, shortness of breath, headache, muscle pains, skin rashes, nausea, chills, eye soreness, diarrhoea, and confusion. Unspecified symptoms could be added as free text.

The symptom questions were updated on November 4, 2020. The original question related to loss of smell and/or taste was then branched into (a) loss of smell and/or taste, and (b) altered smell and/or taste. This update was made to improve the specificity with which symptom severity was assessed. Several other

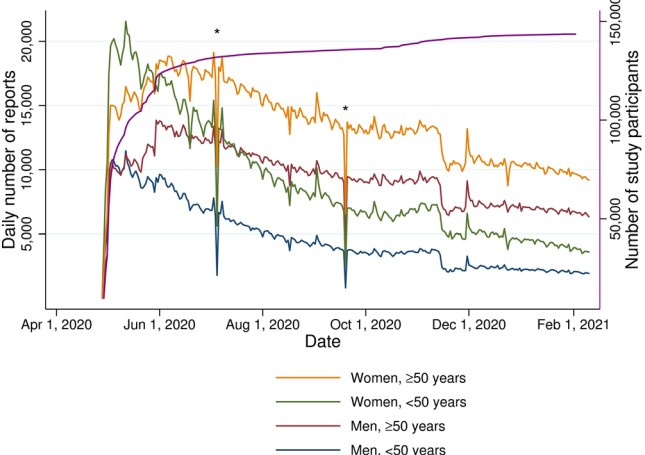

**Fig. 6 Study participation.** Number of daily reports from study participants stratified by sex and age (<50 and ≥50 years), and cumulative number of study participants (n total = 143,531, purple line), in COVID Symptom Study Sweden during the study period April 29, 2020 to February 10, 2021. *Temporary halt in data collection due to technical issue in the COVID Symptom Study app.

new symptom questions were also introduced at this time, including for example unusual hair loss and earache. Questions pertaining to COVID-19 vaccinations were added on March 27, 2021, and are not included in the current analysis. Information on all questionnaire variables both from the baseline health survey and the daily report, including information on whether the questions were available at launch, or the date when the questions were added and/or removed, are presented in Supplementary Table 7.

**Study population and data management.** In this study, we included data from April 29, 2020 to February 10, 2021. Participants were in this study excluded from the analyses if they: (1) never submitted a daily report (n = 5931), (2) had missing age or reported an age <18 years or >99 years (n = 801), or (3) stated their sex as other or intersex (n = 236) as this sample size was insufficient for a further analysis (Supplementary Table 8). Participants whose last report was within 7 days of joining the study (n = 45,483) or did not provide a valid postal code (n = 7310) were excluded from the prevalence estimation, but included in model training if they reported a PCR test and submitted at least one symptomatic daily report within 7 days preceding or on the test date (n = 967). The final study population consisted of 143,531 individuals (Fig. 6). We labelled self-reported height, weight, and body mass index (BMI) as "missing" if <130 or >210 cm, <35 or >300 kg, and <15 or >70 kg/m$^2$, respectively. "Obesity" was defined as BMI ≥30kg/m$^2$. Additional detailed information on data cleaning and pre-processing is available in the Supplementary Material.

**Symptom trajectories in study participants with negative and positive COVID-19 PCR tests.** We described symptom prevalence and trajectories in all CSSS participants from the perspective of their positive (n = 5178) or negative (n = 32,089) PCR test. We included all symptoms reported in the 15 days preceding or following the test date. If a participant reported multiple PCR tests during the study period, only one randomly selected test was included.

**Representativeness of the study population.** We assessed the representativeness of the CSSS study population as a sample of the Swedish general population. We obtained data from the Living Condition Surveys in Sweden from 2018 to 2019 on prevalence of current smoking and BMI in the general adult population[29]. We further extracted aggregate adult population sociodemographic information for all postal code areas in Sweden from Statistics Sweden, which included proportion of women, proportion of inhabitants ≥65 years, highest achieved education level, proportion engaged in work or studies in the population 20–64 years, median yearly net income for the population ≥20 years, and proportion of population ≥18 years with foreign background. Foreign background is here in defined as birth in a country other than Sweden, and/or birth in Sweden but with both parents born in a country other than Sweden. We then calculated a neighborhood deprivation index (NDI, lowest corresponds to most disadvantaged) for each postal code area based on the proportion of adult inhabitants employed or studying, the proportion with a university education, and the median yearly net income[30]. We also calculated population density for all postal code areas in Sweden as the total number of inhabitants per square kilometers. The surface area of each postal code was estimated using shapefiles originating from an external company, Postnummerservice

Norden AB, that contained detailed information on five-digit postal code area boundaries.

**NOVUS**. We compared overall daily symptom prevalence in CSSS during the study period with symptom data collected by NOVUS, a private company that conducts opinion polls and other surveys using panels recruited by random sampling of the Swedish population[31]. Since March 2020, NOVUS has carried out repeated surveys on COVID-19-related symptoms and healthcare contacts, not including PCR test results, with a response rate of ~70%[32]. While in the CSSS participants report symptoms on the same day they experience them, NOVUS participants report any symptoms experienced over the past 14 days even if these reflect their baseline health status (for details, see Supplementary Material).

**Statistical analyses strategy**. Our analysis strategy consisted of five steps, as illustrated in Fig. 2. Each step is detailed below.

Step 1. Training of model for estimation of individual probability of COVID-19

We trained a model to estimate the individual probability of a symptomatic COVID-19, defined as having symptoms and a positive PCR test result. The model training set consisted of data from 19,161 participants who reported at least one PCR test result (of which 2588 were positive) between April 29 and December 31, 2020, and who reported at least one reported candidate symptom within 7 days before or on the test date. As we observed that a higher proportion of study participants reported symptoms within the first week of joining the study than thereafter (Supplementary Fig. 8), we excluded all reports submitted during the first week to reduce participation bias from increased motivation among symptomatic individuals in the general population. For participants who had not submitted reports every day, we assumed the last reported observation to be valid for no more than seven subsequent days. If a participant submitted more than one report on a given day, all reports were combined into a single daily report; a symptom was treated as reported if it was mentioned in at least one of these reports.

We used an L1-penalized logistic regression model (LASSO) to select variables predicting symptomatic COVID-19. The starting set of predictors included all symptoms introduced through May 7, 2020 (excluding hay fever and chills or shivers), as well as their interaction with loss of smell and/or taste, as the latter constituted the strongest predictor of COVID-19[9]. Predictors in the final model were: fever, persistent cough, diarrhoea, delirium, skipped meals, abdominal pain, chest pain, hoarse voice, loss of smell and/or taste, headache, eye soreness, nausea, dizzy or lightheaded, red welts on face or lips, blisters on feet, sore throat, unusual muscle pains, fatigue (mild or severe), and shortness of breath (significant or severe), interaction terms between 14 of those and loss of smell and/or taste, as well as age and sex (see Supplementary Table 9 for model coefficients). After the loss of smell and/or taste symptom question was branched (November 4, 2020) into loss of smell and/or taste and altered smell and/or taste, we combined these two new questions into a single variable, which we subsequently used interchangeably with the original loss of smell and/or taste variable.

As the primary aim was to provide individual probabilities of symptomatic COVID-19, rather than creating a classification rule, we followed the TRIPOD explanation and elaboration document (https://www.equator-network.org/reporting-guidelines/tripod-statement/) guidelines to report discrimination and calibration for the model. Discrimination and calibration were internally evaluated by applying nested tenfold cross-validation within the dataset from April 29 to December 31, 2020. We further assessed calibration and discrimination in CSSS data (1753 participants, of which 339 tested positive) from January 1 to February 10, 2021, the period not included in model training. The nested tenfold cross-validation is described in detail in the Supplementary Material. Discrimination was quantified using the ROC area under the sensitivity-specificity curve (AUC) and the area under the precision-recall curve PR(AUC). The confidence interval for AUC was calculated using the DeLong method[33] and the confidence interval for (PR)AUC by using bootstrap. Model calibration was assessed by plots with estimated probabilities divided into deciles.

To externally validate the model, we used data from the CRUSH Covid study, which invited all individuals (≥18 years) to complete a symptom survey when they did a COVID-19 PCR test in Region Uppsala, the fifth largest healthcare region in Sweden. Using data from October 18, 2020 to February 10, 2021, the classification ability of the model for individual probability of symptomatic COVID-19 was assessed using ROC analysis among individuals who had completed the survey on the day of the test, reported at least one symptom, and had a conclusive test result ($n = 943$; see Supplementary Material and Supplementary Table 10).

Step 2. Estimation of daily individual probability of COVID-19 in CSSS

We used the model from Step 1 to estimate the daily individual probability of symptomatic COVID-19 in the full CSSS study population, including individuals not reporting any PCR test, across the entire study period, from May 10, 2020, to February 10, 2021. Participants not reporting any of the symptoms included in the prediction model were assigned a probability of zero. Participants with long-lasting COVID-19 symptoms were excluded after their 30th day of reporting loss of smell and/or taste to ensure that the estimates were not inflated due to post-acute sequelae of COVID-19. As in Step 1, we (a) excluded symptoms reported during the first 7 days after a participant had joined the study, (b) assumed the last report

to be current for no more than 7 days, and (c) combined all reports on a given day into a single report.

Step 3. Daily regional age- and sex-weighted COVID-19 prevalence estimates

We estimated the daily regional prevalence of symptomatic COVID-19 infection in real-time using a weighted mean of individual predicted probabilities for each of the 21 administrative healthcare regions in Sweden re-weighted by age (<50 and ≥50 years) and sex (women and men) in the total adult population using direct standardization. Demographic data from 2021 was available from Statistics Sweden. The 95% confidence intervals (95% CI) for predictions were generated using the function *ageadjust.direct* from the epitools package in R (version 3.6.1)[34], using the method of Fay and Feuer[35]. This function accommodates the sum of the model-generated probabilities, number of participants for each of the four age- and sex strata on a given day, and the total population of Sweden. The method assumes that the sum of the model-generated probabilities is Poisson-distributed using an approximation based on the gamma distribution.

The odds ratios for all variables in the prediction model were assumed to be generalizable to the background population. Because the model was trained in a dataset with higher prevalence of COVID-19 compared to the general population, the intercept was inflated. We therefore recalibrated the intercept of the model generated in Step 1 until the nationwide app-based predictions for May 27, 2020, matched the estimated nationwide prevalence of 0.3% (95% CI 0.1–0.5%) from a survey performed by the Public Health Agency of Sweden between May 25–28, 2020[11]. In that survey, self-sampling nasal and throat swabs with saliva samples were delivered to a random sample of 2957 individuals (details provided in the Supplementary Material). We assumed both the sensitivity and the proportion of symptomatic COVID-19 in the Public Health Agency of Sweden survey to be 70%[36,37]. Although the method of Fay and Feuer[35] used to calculate the confidence intervals may be regarded as conservative, we assumed the Public Health Agency of Sweden point estimate for May 25–28, 2020, PCR sensitivity, and proportion of asymptomatic individuals to be known quantities.

To compare CSSS prevalence estimates with reported confirmed cases in Sweden, we extracted a pseudonymised dataset of all COVID-19 cases ≥18 in individuals ≥18 years from SmiNet. SmiNet is an electronic notification system of communicable diseases maintained by the Public Health Agency of Sweden, to which all PCR-confirmed cases of COVID-19 are by law reported (details provided in the Supplementary Material). We also acquired data on all COVID-19 hospital admissions in Sweden from the National Patient Register from January 1, 2020 to January 4, 2021, which included all individuals ≥18 years hospitalized with a first diagnosis of COVID-19: International Statistical Classification of Diseases and Related Health Problems Tenth Revision [ICD-10] codes U07.1 and U07.2 (for details see Supplementary Material). The delay in registering COVID-19 hospital admissions in the National Patient Register was up to one month. We therefore utilized data from the register until December 4, 2020. We evaluated the agreement of (a) CSSS-estimated prevalence (main model) (b) CSSS-estimated prevalence (time-dependent model) with official case notification rates from SmiNet and daily new hospital admissions per 100,000 inhabitants ≥18 years (seven-day moving average) on national and regional levels by inspecting trend plots.

We observed a peak in app-based COVID-19 prevalence estimates in mid-September 2020 with no corresponding peak in any disease-specific COVID-19 national register data. However, this peak coincided with regional reports of rhinovirus surges. To better estimate the time course of symptomatic COVID-19 retrospectively from the app data, we constructed a time-dependent model for individual probability of symptomatic COVID-19, based on the variables utilized in Step 1 and with data from the same time period (from April 29 to December 31, 2020) with the addition of restricted cubic splines for calendar time with six knots placed according to Harrell's recommendations[38] (coefficients in Supplementary Table 11). Discrimination and calibration were assessed in CSSS data from January 1 to February 10, 2021, which is a period that was not included in model training. Consistent with the main model, we also performed nested tenfold cross-validation, described in detail in the Supplementary Material, and independently validated the model in the CRUSH Covid dataset.

Step 4. Predictions of regional COVID-19 hospital admissions the following week

We assessed the ability of the CSSS prevalence estimates to predict the following week's COVID-19 hospitalizations during the first and second pandemic waves in Sweden. We defined the end of the first pandemic wave as when the rate of daily new hospital admissions in Sweden dropped below 0.5 individuals per 100,000 inhabitants ≥18 years (July 3, 2020), and the beginning of the second wave as when the hospital admission rate again rose above that threshold (October 19, 2020).

We utilized data from the entire study period in Swedish data to assess which variables to include in the model predicting hospital admission 7 days ahead. The candidate variables were: daily regional CSSS prevalence estimate on day 0 or −1; current regional rate of COVID-19 hospitalizations per 100,000 inhabitants ≥18 years on day 0 or −1; weekday of hospitalization (Monday through Sunday); mean age in region; and mean regional Neighbourhood Deprivation Index (NDI). The final model was a weighted linear regression model, including daily regional CSSS prevalence estimate on day 0 and daily regional rate of COVID-19 hospitalizations per 100,000 inhabitants ≥18 years on day 0, assuming both variables to have a linear relationship with the outcome. Mean age and mean NDI did not notably

influence the predictions and were therefore not included. To create weights for the linear regression we multiplied the number of inhabitants ≥18 years in each region with a density assigned to each day. The density was created using a Gaussian kernel function, with the highest density for the most recent observation. The standard deviation for the kernel function was set to be 2 days by trial-and-error. Weights for days more recent than 7 days prior to the observation being forecasted was set to zero.

The model can be portrayed with the following equation, where $\beta_0$ represents the intercept, "prevalence" the CSSS prevalence and "hospital" the admissions/100,000 inhabitants ≥18 years. The subscript $t_0$ represents day 0 and the subscript $t_7$ day 7.

*Equation 1.*

$$\text{hospital}\_t_7 = \beta_0 + \beta_{prevalence} \times \text{prevalence}\_t_0 + \beta_{hospital} \times \text{hospital}\_t_0$$

We developed an iterative time-updated prediction model process by first training the model in available outcome data up to the first day 0 (June 1, 2020) to derive the coefficients $\beta_0$, $\beta_{prevalence}$ and $\beta_{hospital}$ for Eq. 1. We inserted the CSSS prevalence estimate and daily regional rate of COVID-19 hospitalizations per 100,000 inhabitants ≥18 years on June 1, 2020 into the equation, applying the derived coefficients to predict hospitalization rates on June 8, 2020. We then repeated the model fit and prediction from June 2 to November 29, 2020, with past data influencing the daily new intercept as well as the daily new two betas.

To evaluate the prediction model performance, we transformed both observed and predicted rates to number of hospitalizations per day and region. We then calculated the MdAPE between predicted and observed number of hospitalizations across the first wave (June 8–July 3, 2020), the summer period (July 4–October 18, 2020), and the second wave (October 19–November 29, 2020), for each of the 21 healthcare regions as well as for the five most populated regions combined (Stockholm, Västra Götaland, Skåne, Östergötland and Uppsala). If a region had zero new hospitalizations on any day, that data point was excluded when calculating the absolute percentage error.

For comparison, we further created a second regression model, where we instead of the regional CSSS prevalence estimates inserted the daily regional case notification rates as registered in SmiNet (on day 0) in the equation. We evaluated both date of PCR test and date of registration in SmiNet in the model. When using date of PCR test, we only counted tests that were registered before or on day 0 in each time-updated iteration. Using the date of registration provided better predictions. We found the same density and weights suitable for the SmiNet model as for the CSSS-based model. Model fit was repeated daily across the same period. We further plotted the relative error in % per day and region in hospital admission predictions over the range of observed admissions.

Step 5. Validation of hospital prediction model using external data from England by repeating Steps 2–4

We repeated Step 2 in English data by applying the Swedish model for estimation of individual probabilities from Step 1, applying the same symptom coefficients as in Sweden, and thereby assessed the daily individual probability of symptomatic COVID-19 in ZOE COVID Study (CSS UK) English participants ≥18 years from March 30, 2020 to January 31, 2021. We included participants residing in all postal code areas ($n = 2261$) within any of the seven English healthcare regions (South East, London, North West, East of England, South West, West and East Midlands, Yorkshire and the Humber and North East; total population ≥18 years $n = 43.2$ million). If postal code areas overlapped two or more regions the postal code area was randomly assigned to one of them in our analyses.

We further repeated Step 3 and assessed daily age- and sex-weighted averages of the individual probabilities to estimate daily COVID-19 prevalence across the seven English healthcare regions. Demographic data was extracted from the UK Office for National Statistics[39].

We concluded by seeking replication of Step 4 by applying Eq. 1 in the English dataset using the equivalent variables used in the analysis of CSSS data. We trained the model in available outcome data up to the first day 0 (April 27, 2020) to derive the coefficients $\beta_0$, $\beta_{prevalence}$, and $\beta_{hospital}$. We inserted the ZOE COVID Study prevalence estimate and daily regional rate of COVID-19 hospitalizations per 100,000 inhabitants ≥18 years on April 27, 2020 into the equation, applying the derived coefficients to predict hospitalization rates on May 4, 2020. We then repeated the model fit and prediction from May 5, 2020 to February 11, 2021, with past data influencing the daily new intercept as well as the daily new two betas.

Information on COVID-19 hospital admissions from April 6, 2020 to February 7, 2021, was obtained from the UK government COVID-19 dashboard[40]. We defined the end of the first wave (June 19, 2020), and the start of the second (September 20, 2020), using the same threshold as in Swedish data. We calculated the MdAPEs between predicted and observed number of hospitalizations for each of the seven English healthcare regions during the first wave (May 4–June 19, 2020), the summer period (June 20–September 19, 2020), and the second wave (September 20, 2020–February 7, 2021).

**Reporting summary**. Further information on research design is available in the Nature Research Reporting Summary linked to this article.

## Data availability

Primary data in this study were collected by ZOE Limited and provided to CSSS under a data-sharing agreement. Additional anonymized data originated from the National Board of Health and Welfare, the Public Health Agency of Sweden, Statistics Sweden, and NOVUS. Restrictions apply to the availability of these data, which were used under license and ethical approval and are not publicly available. Pseudonymized individual-level data are, however, available from the authors with written permission from the Swedish Ethical Review Authority (comprehensive information and digital application portal available at https://etikprovningsmyndigheten.se/). Overall, these data may only be used for research, and are not available for commercial use. All applications to the Swedish Ethical Review Authority that involve research on sensitive personal health data must detail scientific purpose, objectives, methods, timetable, data management, ethical considerations, financial matters and competing interests, and also include a project plan and information about the entity responsible for the research as well as the principal investigator. Data requests will be processed within 1 month if a written ethical approval is submitted to the corresponding author. Data delivery is further subject to legal contracts regarding General Data Protection Regulation (GDPR) and Personal Data Processing Agreements between Lund University and/or Uppsala University and the receiving entity.

Group-level prevalence estimates calculated from individual-level data, hospitalization data, and cases of COVID positivity are available as the Source Data (Figures.zip), which were used to generate Figs. 2–5. We provide mock CSSS individual level datasets, that can be used together with the code. These data do not represent real observations. The Source Data that support the findings of this study and mock datasets are available on GitHub at https://github.com/ulfha881/App-based-COVID-19-syndromic-surveillance-and-prediction-of-hospital-admissions-The-COVID-Symptom-S, or, in archived form, at[41].

## Code availability

All code necessary for the replication of our results, including reproducibility instructions, is available on GitHub at https://github.com/ulfha881/App-based-COVID-19-syndromic-surveillance-and-prediction-of-hospital-admissions-The-COVID-Symptom-S, or, in archived form, at[41]. R and Stata codes are provided that will allow readers to generate Figs. 2–5 from Source Data (Figures.zip). Codes to calculate prevalence estimates and the prediction models are provided, but without unmodified individual level datasets, they can't be executed in full.

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

## Acknowledgements

We thank all COVID Symptom Study Sweden participants whose participation, engagement, and feedback were essential to the study. We also thank ZOE Limited for excellent collaboration and development and maintenance of the app. We thank NOVUS for generously sharing their data with us. We thank the CSSS team for their dedication and engagement during the set up and running of the study. In particular, we thank Anna-Maria Dutius Andersson and Mattias Borell for administrative and technical support; Ulrika Blom-Nilsson, Pernilla Siming, and Riia Sustarsic for project management support; Sara Liedholm, Johanna Sandahl, Lars Uhlin, Caroline Runéus, and Katrin Ståhl, along with Lund University and Uppsala University communication teams, for dissemination and outreach. Jacqueline Postma and Lund University and Uppsala University legal teams provided valuable advice regarding the legal aspects of this project; Erik Renström and Stacey Ristinmaa Sörensen also provided valuable advice during the planning phase of the study. Koen Dekkers is thanked for valuable support with computational programming. The computations were enabled by resources in project sens2020559 provided by the Swedish National Infrastructure for Computing (SNIC) at UPPMAX, partially funded by the Swedish Research Council through grant agreement no. 2018-05973.

## Author contributions

M.G., P.F., T.F., B.K., and U.H. conceived the study and designed the analysis. U.H. created and trained the prediction model, and together with H.F., M.M., N.T., and N.O. performed the data analyses. B.K., T.F., and U.H. wrote the first draft of the paper. All authors together interpreted the findings, and reviewed, edited, and approved the final paper. M.G., P.F., and T.F. are the principal investigators, have had full access to all the data in the study, and accept final responsibility for the decision to submit for publication.

## Funding

Swedish Heart-Lung Foundation (20190470, 20140776), Swedish Research Council (EXODIAB, 2009-1039; 2014-03529), European Commission (ERC-2015-CoG - 681742 NASCENT), eSSENCE@LU 8:8 (eSSENCE - The e-Science Collaboration), Swedish Foundation for Strategic Research (LUDC-IRC, 15-0067), Crafoord Foundation (20211011), and EUGLOHRIA (101017752) to M.G. and/or P.F., and European Research Council Starting Grant (ERC-2018-STG - 801965 GUTSY) to T.F. N.O. was financially supported by the Knut and Alice Wallenberg Foundation as part of the National Bioinformatics Infrastructure Sweden at SciLifeLab. A.T.C. was supported in this work through the Massachusetts Consortium on Pathogen Readiness (MassCPR). J.B. was financially supported by Swedish Research Council (2019-00198 and 2021-04665) and by Sweden's Innovation Agency (Vinnova; 2021-02648). ZOE Limited provided in-kind support for all aspects of building, running and supporting the app and service to all users worldwide. Support for this study for KCL researchers was provided by the National Institute for Health Research (NIHR)-funded Biomedical Research Centre based at Guy's and St Thomas' (GSTT) NHS Foundation Trust. This work was also supported by the UK Research and Innovation London Medical Imaging & Artificial Intelligence Centre for Value-Based Healthcare. Investigators also received support from the Wellcome Trust, Medical Research Council (MRC), British Heart Foundation (BHF), Alzheimer's Society, European Union, NIHR, COVID-19 Driver Relief Fund (CDRF) and the NIHR-funded BioResource, Clinical Research Facility and Biomedical Research Centre (BRC) based at GSTT NHS Foundation Trust in partnership with KCL. ZOE Limited developed the app for data collection as a not-for-profit endeavour. None of the funding entities had any role in study design, data analysis, data interpretation, or the writing of this paper. Open access funding provided by Uppsala University.

## Competing interests

CSSS is a strictly non-commercial research project. P.F. consults for and has stock options in ZOE Limited relating to the PREDICT nutrition studies, which are entirely separate from the COVID Symptom Study app development and COVID-19 research. A.T.C. previously served as an investigator on the PREDICT nutrition studies. T.D.S. is a consultant to ZOE Limited. T.S. is the current CEO and a shareholder at Novus

International Group AB, Sweden. R.D., J.W., J.C.P., S.G., A.M., S.S. and J.L.C. work for ZOE Limited. All other authors declare that they have no competing interests.

## Additional information

[1]Department of Medical Sciences, Molecular Epidemiology and Science for Life Laboratory, Uppsala University, Uppsala, Sweden. [2]Genetic and Molecular Epidemiology Unit, Department of Clinical Sciences, Lund University Diabetes Centre, Lund University, Lund, Sweden. [3]Diabetic Complications Unit, Department of Clinical Sciences in Malmö, Lund University Diabetes Centre, Lund, Sweden. [4]Department of Biology, National Bioinformatics Infrastructure Sweden, Science for Life Laboratory, Lund University, Lund, Sweden. [5]Skåne University Hospital, Malmö, Sweden. [6]Clinical Effectiveness Group, Institute of Health and Society, University of Oslo, Oslo, Norway. [7]Department of Medical Epidemiology and Biostatistics, Karolinska Institutet, Stockholm, Sweden. [8]Department of Epidemiology, Harvard T.H. Chan School of Public Health, Boston, MA, USA. [9]Division of Occupational and Environmental Medicine, Lund University, Lund, Sweden. [10]Clinical Studies Sweden, Forum South, Skåne University Hospital, Lund, Sweden. [11]Division of Scientific Computing, Department of Information Technology, Uppsala University, Uppsala, Sweden. [12]Clinical Epidemiology and Biostatistics, School of Medical Sciences, Örebro University, Örebro, Sweden. [13]Integrative Epidemiology, Institute of Environmental Medicine, Karolinska Institutet, Stockholm, Sweden. [14]Biostatistics, School of Public Health and Community Medicine, Institute of Medicine, Sahlgrenska Academy, University of Gothenburg, Gothenburg, Sweden. [15]Department of Public Health and Caring Sciences, Uppsala University, Uppsala, Sweden. [16]Primary Care and Health, Region Uppsala, Uppsala, Sweden. [17]Department of Public Health and Clinical Medicine, Section of Sustainable Health, Umeå University, Umeå, Sweden. [18]Novus Group International AB, Stockholm, Sweden. [19]Department of Health, Medicine and Caring Sciences, Linköping University, Linköping, Sweden. [20]MRC Unit for Lifelong Health and Ageing at UCL, University College London, London, UK. [21]Centre for Medical Image Computing, University College London, London, UK. [22]School of Biomedical Engineering and Imaging Sciences, King's College London, London, UK. [23]ZOE Limited, 164 Westminster Bridge Road, London SE1 7RW, UK. [24]Clinical and Translational Epidemiology Unit, Massachusetts General Hospital and Harvard Medical School, Boston, MA, USA. [25]Department of Twin Research and Genetic Epidemiology, King's College London, London, UK. [26]These authors jointly supervised this work: Maria F. Gomez, Paul W. Franks, Tove Fall. ✉email: tove.fall@medsci.uu.se

