## [Peer Review File · Nature Communications]

Reviewers' Comments:

Reviewer #1:

Remarks to the Author:

This study reports a comprehensive, well-conducted study of COVID-19 positivity and hospitalization monitoring through the CSSS app used by the general population. The data collected were generally consistent with those reported by healthcare organizations in Sweden. The predictions of COVID-19 positivity and hospitalizations are reasonably accurate.

While the study and results are described reasonably well in the manuscript, there are several challenges that should be addressed:

1. Several aspects of the data analysis methodology need to be improved and/or explained better:

a. Figure 1 is too high-level and doesn't explain the data analysis part of the work well. This figure should be expanded, or a new one provided, to explain this part clearly. The main text should also be revised to explain things, such as the time period from which the static and time-dependent models are derived, more clearly. It will also help address several issues indicated below.

b. Since the positive class is much smaller than the negative, AUC is not a reliable measure of performance (see Saito et al, PLoS One, 2015 & Altman et al, Nature Methods, 2021). Class-specific precision, recall and F-measure should also be provided.

c. The methodology for predicting hospitalizations is not clear. Is only the COVID-positive/negative status used as the predictor for this outcome? It is also not clear how the mean error is calculated for these predictions, and what the ground truth is for this evaluation.

d. Since the selected features are determined from a cross-validation on data till December 2020, reporting final results on the same set is invalid, since the training and test sets are the same. These results should be removed from everywhere, and the focus should be on the results on the separate test sets. Another possibility is to conduct nested cross-validation to selected features and evaluate the model (see Whalen et al, Methods, 2016). In either case, the results, as presented currently, are invalid and need to be corrected.

e. Since the data are self-reported, there are likely bias and inaccuracies in them. However, no data cleaning operations and their performance are described. There is also the description of modifications made to the app that would influence data collection and analysis. This should be described more clearly in the main text.

2. Several aspects of the results are also insufficiently explained:

a. It is surprising that the time-dependent model performs equally or worse than the static model, even though the former uses more information. Is this because the model is overfit, the methodology is not strong enough, or something else? There is a comment made in the Discussion that there is a delay in this model, but it is unclear why that would be.

b. The worse performance of the time-dependent model is also confusing, since the authors themselves point out that it matches the SmiNet data better (Figures 4b & 5a).

c. AUC scores of ~ 0.75 don't support the quality of the predictions made by the data and the model. The observed spike in September 2020, as also commented on in the Discussion, also doesn't support the prediction quality. So, the conclusions in the manuscript on the adoptability of the app should be toned down accordingly.

d. The ground truth, not just the predictions, should be included in the plots in Figure 6 and other similar plots in the rest of the paper. This is important for assessing the accuracy of the trends observed.

Addressing these and other reviewers' comments should strengthen the manuscript significantly.

Reviewer #2:

Remarks to the Author:

This a well controlled study to assess whether a symptoms-based App can be useful to predict COVID-19 incidence and hospitalizations. Such information is useful for tracking infection during the pandemic and preparedness for health facilities. In most respects the analyses are well done and the conclusions solid. Limitations of the study are the novelty and minor comments below.

Concerns.

1) A similar set of studies has been published by the UK group using a much larger population (reference 8).

2) The authors could advance the field further by performing more in depth analysis about the effects of a) medication b) ethnicity c) economic status d) comorbidities on hospitalizations. This would add considerable novelty beyond what is already published.

Minor

3) Line 28 Selection of one PCR random test seems problematic. If multiple tests were performed in a short period and initial positive test immediately followed by several negative tests is likely a false positive. Similarly, a negative test follow by several positive test is likely a positive case at least at the downstream tests. I think this could be adjudicated better.

4) Did the authors try other modeling methods besides LASSO?

Reviewer #3:

Remarks to the Author:

The manuscript presents an impressive COVID surveillance effort in Sweden based on self reported information on a digital app.

The researchers develop a model which can reasonably differentiate between users reporting positive vs negative PCR test results based on a range symptoms and some demographic data. The model is developed using cross-validation methods, tested on a period not included in the training data, and then tested again on external data from the CRUSH survey in Sweden.

The researchers then use the model to estimate and predict prevalence of COVID cases and hospitalizations across space and time in Sweden. After initial calibration to the FoHM survey and adjustment to differences in demographics, the researchers report various metrics showing correlation and agreement between the model-based estimates and reported cases and hospitalizations.

The work is impressive in scope. The methods are overall very transparent and clear regarding the methodological choices, and the study employs several internal and external methods to validate the results.

The description regarding the inclusion/exclusion of participants reporting symptoms and PCR tests within a week of joining was unclear and seemed contradictory.

The concepts proposed by the study, whereby self reported symptoms can be used to identify disease in the individual, and to help support surveillance and monitoring of infectious diseases in the population are not original, and have been explored by dozens of studies both prior to COVID and during COVID.

For instance this systematic review summarizes over 169 studies suggesting models to predict

disease and severity in the individual (<https://www.bmj.com/content/369/bmj.m1328.long>), this systematic review surveys over 750 studies on use of digital technologies for disease surveillance (<https://www.nature.com/articles/s41746-021-00407-6>), and these are notable examples for COVID (<https://www.nature.com/articles/s41591-020-0916-2>, <https://www.science.org/doi/full/10.1126/science.abc0473>, <https://www.nature.com/articles/s41591-020-0857-9>).

The model is shown to be predictive on retrospective Swedish data in a limited time-frame. As the model is clearly tailored to the COVID experience and monitoring systems in Sweden, I don't believe the authors provide evidence or rationale which would lead one to expect the model would be generalizable outside of Sweden. It is doubtful performance would remain consistent in Sweden itself. Indeed the authors note the model displayed a non-existent COVID "wave", and therefore provide a more elaborate time-dependent model, which can only be fit to the data retroactively.

Point by point response to reviewers

RE: App-based COVID-19 syndromic surveillance and prediction of hospital admissions: The COVID Symptom Study Sweden (NCOMMS-21-33106). Kennedy et al.

REVIEWERS' COMMENTS

Reviewer #1

This study reports a comprehensive, well-conducted study of COVID-19 positivity and hospitalization monitoring through the CSSS app used by the general population. The data collected were generally consistent with those reported by healthcare organizations in Sweden. The predictions of COVID-19 positivity and hospitalizations are reasonably accurate.

While the study and results are described reasonably well in the manuscript, there are several challenges that should be addressed:

Several aspects of the data analysis methodology need to be improved and/or explained better:

Comment 1a. Figure 1 is too high-level and doesn't explain the data analysis part of the work well. This figure should be expanded, or a new one provided, to explain this part clearly. The main text should also be revised to explain things, such as the time period from which the static and time-dependent models are derived, more clearly. It will also help address several issues indicated below.

Authors' reply: Thank you for these constructive comments. We have now generated an expanded and more granular Figure to better explain each of the main steps of the analyses. We have also edited the Results and Methods sections to be consistent with the five steps shown in the Figure (renumbered to Figure 2). The revised text now includes a clearer description of the time periods for the static and time-dependent models, as well as additional edits that aim to clarify and ensure adherence to *Nature Communications* formatting rules.

New Figure 2, on page 37:

Figure 2. Analysis strategy and data sources

Previous Figure 1.

New sections in the Results, on pages 7-10:

“Step 1. Training of model for estimation of individual probability of COVID-19

Our analysis strategy consisted of five steps, as illustrated in Figure 2. In the first step, we developed a model to estimate individual probability of symptomatic COVID-19, utilizing information from 19,161 CSSS participants who reported at least one PCR test

(of whom 2,586 were COVID-19 infection positive) between April 29 and December 31, 2020; these individuals also reported at least one candidate symptom within seven days before or on the test date. The final model selected by LASSO included 17 symptoms and sex, as well as 2-way interactions between loss of smell and/or taste and 14 symptoms, and a 2-way interaction between loss of smell and/or taste and sex. The AUC for this main model was 0.76 (95% CI 0.75–0.78) during the training period April 29 to December 31, 2020; n=19,161) and 0.72 (95% CI 0.69–0.75) during the evaluation period (January 1 to February 10, 2021; n=1,753) and 0.78 (95% CI 0.74–0.83) in the external dataset of 943 symptomatic individuals from the CRUSH Covid survey (144 positive; test positivity 15.3%; October 18, 2020 to February 10, 2021). Calibration graphs are available in Supplementary Figure 3.

Step 2. Estimation of daily individual probability of COVID-19 in CSSS

We applied the model from Step 1 to estimate the daily individual probability of symptomatic COVID-19 in all CSSS study participants, including non-tested individuals, across the entire study period from May 10, 2020, through February 10, 2021. Non-symptomatic individuals were assigned a probability of 0 for that day.

Step 3. Daily regional COVID-19 prevalence estimates in the general population

The individual probabilities from Step 2 were then used to estimate the daily regional COVID-19 prevalence in the general population in Sweden, accounting for differences in age and sex distributions of the participants as compared to the general population in each region. We calibrated the intercept of the model generated in step 1 so that the estimated prevalence of May 27, 2020 matched the estimated prevalence from a national COVID-19 prevalence survey (n=2,957). The resulting CSSS prevalence estimates of symptomatic COVID-19 showed similar waves as the first and second waves of COVID-19 hospitalization (Figure 3a, Supplementary Figure 4a). In contrast, data from SmiNet, the national Swedish register on laboratory-confirmed cases of COVID-19, did not capture the first wave (Figure 3b, Supplementary Figure 4b).

[...]

Step 4. Prediction of regional COVID-19 hospital admissions the following week

We developed an iterative time-updated prediction model to assess if the regional prevalence estimates from Step 3 could be used together with current hospital data for prediction of regional COVID-19 hospital admissions seven days ahead. The

parameters were estimated on available data through June 1, 2020 with larger weights applied to more recent observations, to predict admissions on June 8, 2020. This procedure was repeated to calibrate the coefficients throughout the study period. Weights and model specifications were based on exploratory analyses using data from May 11 to November 29, 2020. Overall, 16,752 individuals (≥ 18 years) were admitted to hospital with a diagnosis of COVID-19 from May 11 through November 29, 2020, and the number of daily new COVID-19 hospital admissions ranged from 0–104 across the 21 Swedish healthcare regions.

[...]

Step 5. Validation of the hospitalization prediction model in England

We sought to validate the CSSS-based hospitalization prediction model in England by repeating Steps 2 and 3 and parts of Step 4. The English dataset encompassed daily reports from 2,638,536 ZOE COVID Study (CSS UK) English study participants from March 30, 2020 to January 31, 2021 (study population characteristics and regional participation rates are available in Supplementary Tables 3 and 4). We extracted information on all COVID-19 hospital admissions ($n=318,232$) in individuals ≥ 18 years across the seven English healthcare regions from April 6, 2020 through February 7, 2021 from National Health Service England data. The number of new daily COVID-19 hospital admissions ranged from 0–958 across the English regions during this period.

We applied the exact same model that was developed in Step 1 in CSSS to estimate daily individual probability of COVID-19 in the English dataset, and then estimated the daily age- and sex-weighted COVID-19 prevalence across the English regions. We further applied the same iterative time-updated prediction model as in the Swedish dataset to predict hospital admissions the following week in the seven English regions. We used available outcome data up to April 27, 2020 to tune the parameters and to predict admissions a week later on May 4, 2020 and then repeated this daily throughout the study period (until February 7, 2021). [...]"

Updated information on time periods in the Methods, on pages 19-20 and 22-23:

“Step 1. Training of model for estimation of individual probability of COVID-19

We trained a model to estimate the individual probability of a symptomatic COVID-19, defined as having symptoms and a positive PCR test result. The model training set consisted of data from 19,161 participants who reported at least one PCR test result (of which 2,588 were positive) between April 29 and December 31, 2020, and who reported at least one reported candidate symptom within seven days before or on the test date.”

and

“As the primary aim was to provide individual probabilities of symptomatic COVID-19, rather than creating a classification rule, we followed the TRIPOD explanation and elaboration document (<https://www.equator-network.org/reporting-guidelines/tripod-statement/>) guidelines to report discrimination and calibration for the model. Discrimination and calibration were internally evaluated by applying nested tenfold cross-validation within the dataset from April 29 to December 31, 2020. We further assessed discrimination and calibration in CSSS data (1,753 participants, of which 339 tested positive) from January 1 to February 10, 2021, the period not included in model training. The nested tenfold cross-validation is described in detail in the Supplementary Material. Discrimination was quantified using the ROC area under the curve (AUC) as this metric is robust to differences in prevalence. Model calibration was assessed by plots with estimated probabilities divided into deciles.”

and

“We observed a peak in app-based COVID-19 prevalence estimates in mid-September 2020 with no corresponding peak in any disease-specific COVID-19 national register data. However, this peak coincided with regional reports of rhinovirus surges. To better estimate the time course of symptomatic COVID-19 retrospectively from the app data, we constructed a time-dependent model for individual probability of symptomatic COVID-19, based on the variables utilized in Step 1 and with data from the same time period (from April 29 to December 31, 2020) with the addition of restricted cubic splines for calendar time with six knots placed according to Harrell's recommendations² (coefficients in Supplementary Table 10). Discrimination and calibration were assessed in CSSS data from January 1 to February 10, 2021, which is a period that was not included in model training. Consistent with the main model, we also performed nested tenfold cross-validation, described in detail in the

Supplementary Material, and independently validated the model in the CRUSH Covid dataset.”

Comment 1b: Since the positive class is much smaller than the negative, AUC is not a reliable measure of performance (see Saito et al, PLoS One, 2015 & Altman et al, Nature Methods, 2021). Class-specific precision, recall and F-measure should also be provided.

Authors' reply: We wish to thank the Reviewer for the opportunity to clarify the aim of our model and the choice of evaluation metrics.

The two articles supplied by the Reviewer discuss the scenario when the aim of a model is binary classification, a case where we fully agree that good precision/positive predictive value is of great importance. Indeed, if our aim had been to use the model in this way, classifying study participants as having either symptomatic COVID-19 or not, our model would work rather poorly, despite a moderately high AUC. After recalibrating the intercept (described on page 21 in the manuscript) the model very rarely generated COVID-19-probabilities above 0.3. This means that a cut-off of 0.5 would assign basically no one to a COVID-19-positive status. And with a lower threshold, assuming that the generated probabilities are approximately correct, a clear majority classified as COVID-19-symptomatic would in fact be false positives.

However, the aim of the current study was not to classify but to estimate nationwide and regional prevalence of symptomatic COVID-19 to aid hospital admission prediction. This estimation was done by assigning each study participant a probability of symptomatic COVID-19 given their age, sex, and symptomology, setting all non-symptomatic to 0. We then took a demography-weighted average of these probabilities to provide an estimate of the prevalence.

For this to be feasible, we needed the model to be able to 1) discriminate reasonably well as evaluated by the AUC, and 2) to generate reasonable COVID-19-probabilities as evaluated by calibration curves. In our opinion, these are the two important metrics for us. We also refer to the TRIPOD: explanation and elaboration document (<https://www.equator-network.org/reporting-guidelines/tripod-statement/>), which deals with classical statistical predictive models in which the aim is to generate probabilities of an outcome rather than create a classification rule. There, positive predictive values are not discussed, only discrimination and calibration evaluation.

For transparency, we supply the area under the precision-recall-curve (PRC(AUC)), mentioned in the article by Saito et al. in this response letter. If the Reviewer and/or Editor desires, we can include this in the manuscript, but we want to stress that these values are actually inflated, as the CRUSH Covid validation sample used for evaluating performance is a dataset of COVID PCR tests, in which the positive class is much more prevalent than it is in the general population. We argue that regular AUC,

due to not being sensitive to prevalence, is more generalizable than PRC(AUC). We cannot provide precision, recall and F-measure, since we do not use any cut-off values. Please note that the PRC needs to be interpreted together with the prevalence, as a random classifier would yield the same PRC as the prevalence.

Table A. PRC(AUC) (with bootstrap-based 95% CIs)

	Main model	Time-dependent model
Training data	0.38 (95% CI 0.35–0.40)	0.49 (95% CI 0.46–0.50)
Test data	0.40 (95% CI 0.35–0.45)	0.38 (95% CI 0.33–0.44)
CRUSH Covid	0.48 (95% CI 0.40–0.56)	0.35 (95% CI 0.27–0.41)

We have clarified our approach in the Methods and in the Supplementary Material.

Updated section of the Methods section, on pages 20 and 22-23:

“As the primary aim was to provide individual probabilities of symptomatic COVID-19, rather than creating a classification rule, we followed the TRIPOD explanation and elaboration document (<https://www.equator-network.org/reporting-guidelines/tripod-statement/>) guidelines to report discrimination and calibration for the model. Discrimination and calibration were internally evaluated by applying nested tenfold cross-validation within the dataset from April 29 to December 31, 2020. We further assessed discrimination and calibration in CSSS data (1,753 participants, of which 339 tested positive) from January 1 to February 10, 2021, the period not included in model training. The nested tenfold cross-validation is described in detail in the Supplementary Material. Discrimination was quantified using the ROC area under the curve (AUC) as this metric is robust to differences in prevalence. Model calibration was assessed by plots with estimated probabilities divided into deciles.”

and

“We observed a peak in app-based COVID-19 prevalence estimates in mid-September 2020 with no corresponding peak in any disease-specific COVID-19 national register data. However, this peak coincided with regional reports of rhinovirus surges. To better estimate the time course of symptomatic COVID-19 retrospectively from the app data, we constructed a time-dependent model for individual probability of symptomatic COVID-19, based on the variables utilized in Step 1 and with data from the same time period (from April 29 to December 31, 2020) with the addition of

restricted cubic splines for calendar time with six knots placed according to Harrell's recommendations² (coefficients in Supplementary Table 10). Discrimination and calibration were assessed in CSSS data from January 1 to February 10, 2021, which is a period that was not included in model training. Consistent with the main model, we also performed nested tenfold cross-validation, described in detail in the Supplementary Material, and independently validated the model in the CRUSH Covid dataset.”

Comment 1c: The methodology for predicting hospitalizations is not clear. Is only the COVID-positive/negative status used as the predictor for this outcome? It is also not clear how the mean error is calculated for these predictions, and what the ground truth is for this evaluation.

Authors' reply: Thank you. We agree that it was not clear what data we included in the hospitalization prediction model or how we calculated the mean errors. In this revised version, we have re-worked the hospital admission model to an iterative time-updated prediction model of daily admissions seven days ahead, based on current regional app-based prevalence and current admissions. The model can be portrayed with the following equation, where β_0 represents the intercept, "prevalence" the CSSS prevalence and "hospital" the admissions/100,000 inhabitants ≥ 18 years. On each day, the model coefficients were fit using available data at that time point with larger weights on larger regions and later observations. The subscript t_0 represents day 0 and the subscript t_7 day 7.

Equation 1.

$$\text{hospital}_{t_7} = \beta_0 + \beta_{\text{prevalence}} \times \text{prevalence}_{t_0} + \beta_{\text{hospital}} \times \text{hospital}_{t_0}$$

We have now thoroughly revised the text for clarity and transparency. In addition, we now present estimates of median absolute percentage error instead of mean absolute error, in order to better compare regions of varying size and with the validation data set. The errors were calculated transforming both observed and predicted rates to number of hospitalizations per day and region. We then calculated the median absolute percentage error (MdAPE) between predicted and observed number of hospitalizations across the first wave (May 11 to July 3, 2020), the summer period (July 4 to October 18, 2020), and the second wave (October 19 to November 29, 2020), for each of the 21 healthcare regions as well as for the five most populated regions combined (Stockholm, Västra Götaland, Skåne, Östergötland and Uppsala). If a region had zero new hospitalizations on any day, that data point was excluded when calculating the absolute percentage error.

Updated sections of the Methods, on pages 23-26:

"Step 4. Predictions of regional COVID-19 hospital admissions the following week

We assessed the ability of the CSSS prevalence estimates to predict the following week's COVID-19 hospitalizations during the first and second pandemic waves in Sweden. We defined the end of the first pandemic wave as when the rate of daily new

hospital admissions in Sweden dropped below 0.5 individuals per 100,000 inhabitants ≥ 18 years (July 3, 2020), and the beginning of the second wave as when the hospital admission rate again rose above that threshold (October 19, 2020).

We utilized data from the entire study period in Swedish data to assess which variables to include in the model predicting hospital admission seven days ahead. The candidate variables were: daily regional CSSS prevalence estimate on day 0 or -1; current regional rate of COVID-19 hospitalizations per 100,000 inhabitants ≥ 18 years on day 0 or -1; weekday of hospitalization (Monday through Sunday); mean age in region; and mean regional Neighbourhood Deprivation Index (NDI). The final model was a weighted linear regression model, including daily regional CSSS prevalence estimate on day 0 and daily regional rate of COVID-19 hospitalizations per 100,000 inhabitants ≥ 18 years on day 0, assuming both variables to have a linear relationship with the outcome. Mean age and mean NDI did not notably influence the predictions and were therefore not included. To create weights for the linear regression we multiplied the number of inhabitants ≥ 18 years in each region with a density assigned to each day. The density was created using a Gaussian kernel function, with the highest density for the most recent observation. The standard deviation for the kernel function was set to be two days by trial-and-error. Weights for days more recent than seven days prior to the observation being forecasted was set to zero.

The model can be portrayed with the following equation, where β_0 represents the intercept, “prevalence” the CSSS prevalence and “hospital” the admissions/100,000 inhabitants ≥ 18 years. The subscript t_0 represents day 0 and the subscript t_7 day 7.

Equation 1.

$$\text{hospital}_{t_7} = \beta_0 + \beta_{\text{prevalence}} \times \text{prevalence}_{t_0} + \beta_{\text{hospital}} \times \text{hospital}_{t_0}$$

We developed an iterative time-updated prediction model process by first training the model in available outcome data up to the first day 0 (June 1, 2020) to derive the coefficients β_0 , $\beta_{\text{prevalence}}$ and β_{hospital} for Eq. 1. We inserted the CSSS prevalence estimate and daily regional rate of COVID-19 hospitalizations per 100,000 inhabitants ≥ 18 years on June 1, 2020 into the equation, applying the derived coefficients to predict hospitalization rates on June 8, 2020. We then repeated the model fit and prediction from June 2 to November 29, 2020, with past data influencing the daily new intercept as well as the daily new two betas.

To evaluate the prediction model performance, we transformed both observed and predicted rates to number of hospitalizations per day and region. We then calculated the median absolute percentage error (MdAPE) between predicted and observed number of hospitalizations across the first wave (May 11 to July 3, 2020), the summer period (July 4 to October 18, 2020), and the second wave (October 19 to November 29, 2020), for each of the 21 healthcare regions as well as for the five most populated regions combined (Stockholm, Västra Götaland, Skåne, Östergötland and Uppsala). If a region had zero new hospitalizations on any day, that data point was excluded when calculating the absolute percentage error.

For comparison, we further created a second regression model, where we instead of the regional CSSS prevalence estimates inserted the daily regional case notification rates as registered in SmiNet (on day 0) in the equation. We evaluated both date of PCR test and date of registration in SmiNet in the model. When using date of PCR test, we only counted tests that were registered before or on day 0 in each time-updated iteration. Using the date of registration provided better predictions. We found the same density and weights suitable for the SmiNet model as for the CSSS-based model. Model fit was repeated daily across the same period. We further plotted the relative error in % per day and region in hospital admission predictions over the range of observed admissions.

Step 5. Validation of hospital prediction model using external data from England by repeating Steps 2–4

We repeated Step 2 in English data by applying the Swedish model for estimation of individual probabilities from Step 1, applying the same symptom coefficients as in Sweden, and thereby assessed the daily individual probability of symptomatic COVID-19 in ZOE COVID Study (CSS UK) English participants ≥ 18 years from March 30, 2020 to January 31, 2021. We included participants residing in all postal code areas ($n=2,261$) within any of the seven English healthcare regions (South East, London, North West, East of England, South West, West and East Midlands, Yorkshire and the Humber and North East; total population ≥ 18 years $n=43.2$ million). If postal code areas overlapped two or more regions the postal code area was randomly assigned to one of them in our analyses.

We further repeated Step 3 and assessed daily age- and sex-weighted averages of the individual probabilities to estimate daily COVID-19 prevalence across the seven English

healthcare regions. Demographic data was extracted from the UK Office for National Statistics³.

We concluded by seeking replication of Step 4 by applying Eq. 1 in the English dataset using the equivalent variables used in the analysis of CSSS data. We trained the model in available outcome data up to the first day 0 (April 27, 2020) to derive the coefficients β_0 , $\beta_{\text{prevalence}}$, and β_{hospital} . We inserted the ZOE COVID Study prevalence estimate and daily regional rate of COVID-19 hospitalizations per 100,000 inhabitants ≥ 18 years on April 27, 2020 into the equation, applying the derived coefficients to predict hospitalization rates on May 4, 2020. We then repeated the model fit and prediction from May 5, 2020 to February 11, 2021, with past data influencing the daily new intercept as well as the daily new two betas.

Information on COVID-19 hospital admissions from April 6, 2020 to February 7, 2021, was obtained from the UK government COVID-19 dashboard⁴. We defined the end of the first wave (June 19, 2020), and the start of the second (September 20, 2020), using the same threshold as in Swedish data. We calculated the MdAPEs between predicted and observed number of hospitalizations for each of the seven English healthcare regions during the first wave (April 6 to June 19, 2020), the summer period (June 20 to September 19, 2020), and the second wave (September 20, 2020 to February 7, 2021).”

Comment 1d: Since the selected features are determined from a cross-validation on data till December 2020, reporting final results on the same set is invalid, since the training and test sets are the same. These results should be removed from everywhere, and the focus should be on the results on the separate test sets. Another possibility is to conduct nested cross-validation to selected features and evaluate the model (see Whalen et al, Methods, 2016). In either case, the results, as presented currently, are invalid and need to be corrected.

Authors' reply: We thank the Reviewer for pointing out that the description of our cross-validation procedure was unclear and needed further elaboration. We agree with the Reviewer that nested cross-validation is appropriate for evaluating the model on the training data and had in fact already used this procedure applying the steps below. We would like to stress that when evaluating model performance, neither variable selection nor model selection has been performed on the entire dataset – it is all part of the nested cross-validation loop. This was unclear in the initial version and we have now clarified and expanded this section in the manuscript. We performed the following steps (applying terminology from the article by Oppedal et al.⁵):

1. We divide the data into ten different folds. For each iteration, we run a LASSO on nine “training” folds and generate probabilities of symptomatic COVID-19 for the individuals in the 10th test fold. This is the outer loop.
2. The shrinkage parameter for each of the 10 LASSO models is determined by dividing the outer loop training data into 10 additional parts, performing a cross-validation and selecting the shrinkage parameter which minimizes the cross-validation error. This is the inner loop.
3. Each of the ten LASSO models in the outer loop will thus contain different shrinkage parameters, different coefficients, and potentially a different set of predictors.
4. We will end up with a set of predicted “pre-validated” COVID-19 probabilities for all individuals.
5. The AUCs are then calculated by comparing the predicted probabilities to the actual class labels of those individuals.

Updated paragraphs in the Material and Methods, on pages 19, 20, and 22-23:

“We used an L1-penalized logistic regression model (LASSO) to select variables predicting symptomatic COVID-19.”

and

“As the primary aim was to provide individual probabilities of symptomatic COVID-19, rather than creating a classification rule, we followed the TRIPOD explanation and elaboration document (<https://www.equator-network.org/reporting-guidelines/tripod-statement/>) guidelines to report discrimination and calibration for the model. Discrimination and calibration were internally evaluated by applying nested tenfold cross-validation within the dataset from April 29 to December 31, 2020. We further assessed discrimination and calibration in CSSS data (1,753 participants, of which 339 tested positive) from January 1 to February 10, 2021, the period not included in model training. The nested tenfold cross-validation is described in detail in the Supplementary Material. Discrimination was quantified using the ROC area under the curve (AUC) as this metric is robust to differences in prevalence. Model calibration was assessed by plots with estimated probabilities divided into deciles.”

and

“We observed a peak in app-based COVID-19 prevalence estimates in mid-September 2020 with no corresponding peak in any disease-specific COVID-19 national register data. However, this peak coincided with regional reports of rhinovirus surges. To better estimate the time course of symptomatic COVID-19 retrospectively from the app data, we constructed a time-dependent model for individual probability of symptomatic COVID-19, based on the variables utilized in Step 1 and with data from the same time period (from April 29 to December 31, 2020) with the addition of restricted cubic splines for calendar time with six knots placed according to Harrell's recommendations² (coefficients in Supplementary Table 10). Discrimination and calibration were assessed in CSSS data from January 1 to February 10, 2021, which is a period that was not included in model training. Consistent with the main model, we also performed nested tenfold cross-validation, described in detail in the Supplementary Material, and independently validated the model in the CRUSH Covid dataset.”

New section in the Supplementary Material, on pages 4-5:

“The nested tenfold cross-validation of the model developed in Step 1 to estimate individual probability of COVID-19

In both the main model and the time-dependent model, we performed internal evaluation of discrimination and calibration by applying nested ten-fold cross-validation within the dataset from April 29 to December 31, 2020. In a nested ten-fold cross-validation, Oppedal et al⁵ outlined an outer and an inner loop of iterations, see Figure 5 in their paper. For each of ten iterations in the outer loop, we ran a LASSO on nine folds constituting the "training data" and generated symptomatic COVID-19 probabilities for the individuals in the test data (the last fold). The shrinkage parameter for each LASSO model was determined by dividing the outer loop training data into 10 additional parts, performing an inner loop cross-validation, then selecting the shrinkage parameter which minimizes inner loop cross-validation error. Each of the 10 LASSO models in the outer loop will thus contain different shrinkage parameters, different coefficients, and potentially a different set of predictors. We will end up with a set of predicted "pre-validated" COVID-19 probabilities for all individuals. Since the probabilities are generated from different models, the area under the curve (AUC) will be an evaluation of the model-generating process rather than of a specific model."

Comment 1e: Since the data are self-reported, there are likely bias and inaccuracies in them. However, no data cleaning operations and their performance are described. There is also the description of modifications made to the app that would influence data collection and analysis. This should be described more clearly in the main text.

Authors' reply: Thank you for these constructive comments. We have now included the potential limitations of self-reported data in the Discussion. We have also added information in the main manuscript on data cleaning. Additional inclusion and exclusion criteria are now also available under the Methods subheading "Study population and data management".

Updated limitations section in the Discussion, on pages 13-14:

"Additional potential limitations to the data collection also apply. Firstly, participants may have been more likely to join the study and report daily if they experienced symptoms perceived to be linked with COVID-19 than if they had been healthy, potentially inflating COVID-19 prevalence estimates. We sought to reduce this risk by excluding the first seven days of data collected for each participant, but we cannot exclude the risk of residual participation bias. Secondly, all data collected in the COVID Symptom Study app are self-reported. We had no means of linking the self-reported data to national population or health registers, so we could not validate information on COVID-19 tests or baseline health survey questions. We were able to overcome this limitation by validating the model predicting COVID-19 PCR test positivity in the independent CRUSH Covid dataset. Lastly, some questions in the COVID Symptom Study app were updated during the study period, potentially influencing data collection and analyses. The baseline health survey questions that were modified and/or updated during this time were, however, not included in our analyses and did not affect the prevalence estimates or the accuracy of the hospitalization prediction model. The symptom questions were updated on November 4, 2020. This update was implemented at the same time as incidence was increasing sharply, but the national prevalence estimate curve and the regional prevalence estimate curves remained smooth, indicating that these updates did not materially influence our findings."

and

Updated paragraph in Methods, on page 17:

"Study population and data management"

In this study, we included data from April 29, 2020 to February 10, 2021. Participants were in this study excluded from the analyses if they: 1) never submitted a daily report (n=5,931), 2) had missing age or reported an age <18 years or >99 years (n=801), or 3) stated their sex as other or intersex (n=236) as this sample size was insufficient for a further analysis (Supplementary Table 7). Participants whose last report was within seven days of joining the study (n=45,483) or did not provide a valid postal code (n=7,310) were excluded from the prevalence estimation, but included in model training if they reported a PCR test and submitted at least one symptomatic daily report within seven days preceding or on the test date (n=967). The final study population consisted of 143,531 individuals (Figure 6). We labelled self-reported height, weight, and body mass index (BMI) as “missing” if <130 or >210 cm, <35 or >300 kg, and <15 or >70kg/m², respectively. “Obesity” was defined as BMI ≥30kg/m².”

With regards to the COVID Symptom Study app updates, the baseline health survey questions on lung disease and diabetes were updated and branched in late June 2020. Other baseline questions, relating to for example dietary supplements and social distancing, were introduced in June 2020 but subsequently removed from the app in September/October 2020 (detailed information on all question updates is available in Supplementary Table 1). However, such baseline health survey variables are not included in any models or analyses in this study, and these updates do therefore not influence our estimates.

The symptom questions were updated on November 4, 2020. The original question relating to loss of smell and/or taste was then branched into a) loss of smell and/or taste, and b) altered smell and/or taste. The rationale for this update was to add specificity to the severity of the symptoms which may be beneficial in the study of post-acute sequelae of COVID-19 as well as when investigating if symptom presentation might vary across study participation groups. For continuity, we subsequently combined the two questions into a single variable (“loss of/altered smell and/or taste”) in both in the prediction model training (Step 1) and in the daily individual probability of symptomatic COVID-19 (Step 2) used to produce the estimated of daily regional COVID-19 prevalence in the general population (Step 3). This question update was implemented at the same time as incidence was increasing sharply, but the national prevalence estimate curve (Figure 3a and 3b “Main model”) as well as the regional prevalence estimate curves remained smooth (Supplementary Figure 4 “App estimation”), indicating that the question update did not exert any overall large influence on our findings. Other symptom questions that were introduced on November 4, such as unusual hair loss

and earache, were not included in the prediction model training (Step 1) and do therefore not influence our models or estimates. We have now included this information in the manuscript.

Updated section on the ZOE COVID Symptom Study app updates in the Methods, on pages 16-17 and 19-20:

“COVID Symptom Study Sweden (CSSS)

[...]

The symptom questions were updated on November 4, 2020. The original question related to loss of smell and/or taste was then branched into a) loss of smell and/or taste, and b) altered smell and/or taste. This update was made to improve the specificity with which symptom severity was assessed. Several other new symptom questions were also introduced at this time, including for example unusual hair loss and earache. Questions pertaining to COVID-19 vaccinations were added on March 27, 2021, and are not included in the current analysis. Information on all questionnaire variables both from the baseline health survey and the daily report, including information on whether the questions were available at launch, or the date when the questions were added and/or removed, are presented in Supplementary Table 6.”

and

“Step 1. Training of algorithm for estimation of individual probability of COVID-19

[...]

Predictors in the final model were: fever, persistent cough, diarrhoea, delirium, skipped meals, abdominal pain, chest pain, hoarse voice, loss of smell and/or taste, headache, eye soreness, nausea, dizzy or lightheaded, red welts on face or lips, blisters on feet, sore throat, unusual muscle pains, fatigue (mild or severe), and shortness of breath (significant or severe), interaction terms between 14 of those and loss of smell and/or taste, as well as age and sex (see Supplementary Table 8 for model coefficients). After the loss of smell and/or taste symptom question was branched (November 4, 2020) into loss of smell and/or taste and altered smell and/or taste, we combined these two new questions into a single variable, which we subsequently used interchangeably with the original loss of smell and/or taste variable.

and

As the primary aim was to provide individual probabilities of symptomatic COVID-19, rather than creating a classification rule, we followed the TRIPOD explanation and elaboration document (<https://www.equator-network.org/reporting-guidelines/tripod-statement/>) guidelines to report discrimination and calibration for the model. Discrimination and calibration were internally evaluated by applying nested tenfold cross-validation within the dataset from April 29 to December 31, 2020. We further assessed discrimination and calibration in CSSS data (1,753 participants, of which 339 tested positive) from January 1 to February 10, 2021, the period not included in model training. The nested tenfold cross-validation is described in detail in the Supplementary Material. Discrimination was quantified using the ROC area under the curve (AUC) as this metric is robust to differences in prevalence. Model calibration was assessed by plots with estimated probabilities divided into deciles.”

Updated section on limitations in the Discussion, pages 13-14:

“Some additional potential however also apply. [...] Lastly, some questions in the COVID Symptom Study app were updated during the study period, potentially influencing data collection and analyses. The baseline health survey questions that were modified and/or updated during this time were, however, not included in our analyses and did not affect the prevalence estimates or the accuracy of the hospitalization prediction model. The symptom questions were updated on November 4, 2020. This update was implemented at the same time as incidence was increasing sharply, but the national prevalence estimate curve and the regional prevalence estimate curves remained smooth, indicating that these updates did not materially influence our findings.”

Updated section on the COVID Symptom Study app updates in the Supplementary Material, on page 1:

“Question updates in the COVID Symptom Study Sweden baseline survey

Upon downloading the COVID Symptom Study Sweden (CSSS) app and providing informed consent, participants were asked to complete a health survey including pre-existing health conditions. The baseline health survey included questions on diabetes and lung disease that were later expanded to include disease subtypes in late June

2020. Prior to that, participants could only respond yes/no to presence of “diabetes” and “lung disease”. Following the updates, participants who had previously indicated presence of diabetes and/or lung disease could specify type of disease, and new participants were asked the more detailed questions upon registration. Consequently, participants who did not submit a single daily report after June 24, 2020 were all labelled as having “type not specified” diabetes and/or lung disease. Other baseline questions, for example on dietary supplements or physical distancing, were introduced in June 2020 but subsequently removed from the app in September/October 2020. Information on all questionnaire variables both from the baseline health survey and the daily report, including information on whether the question was available at launch, or the date when the question was added and/or removed, are presented in Supplementary Table 6.”

Comment 2: Several aspects of the results are also insufficiently explained:

a. It is surprising that the time-dependent model performs equally or worse than the static model, even though the former uses more information. Is this because the model is overfit, the methodology is not strong enough, or something else? There is a comment made in the Discussion that there is a delay in this model, but it is unclear why that would be.

and

b. The worse performance of the time-dependent model is also confusing, since the authors themselves point out that it matches the SmiNet data better (Figures 4b & 5a).

Authors' reply: Thank you. Our interpretation is that the time-dependent model gives a more accurate view of the community infection rates in the Swedish population than the static model, however, with a lag. The time-dependent model also has a problem with extrapolation. Although COVID-19 prevalence increased during the end of 2020, it is unrealistic to assume that this increasing trend will continue unabated during the entire 2021 validation period (given that other variables in the model are held constant). Yet this is what the time-dependent model assumes.

Furthermore, the time-dependent model is strongly influenced by the proportion of positive COVID-19 tests in the model training dataset during a specific period. The model is thus insensitive to a sudden increase or decrease in the reporting of symptoms, and any upward or downward trend in prevalence will not be fully captured until this increase/decrease is also reflected in the COVID-19 test results in the app. The static model, on the other hand, will respond directly to an increase of reported symptoms, and while increasing the sensitivity of the model, it also increases the risk of false alerts. This is what we mean by "delay" - the time taken from when a COVID-19 symptomatic app user reports new symptoms to the time when the same person reports a positive test result. We have now expanded on these issues in the manuscript.

Updated paragraph in the Methods section, on pages 22-23:

"We observed a peak in app-based COVID-19 prevalence estimates in mid-September 2020 with no corresponding peak in any disease-specific COVID-19 national register data. However, this peak coincided with regional reports of rhinovirus surges. To better estimate the time course of symptomatic COVID-19 retrospectively from the app data, we constructed a time-dependent model for individual probability of symptomatic COVID-19, based on the variables utilized in Step 1 and with data from

the same time period (from April 29 to December 31, 2020) with the addition of restricted cubic splines for calendar time with six knots placed according to Harrell's recommendations² (coefficients in Supplementary Table 10). Discrimination and calibration were assessed in CSSS data from January 1 to February 10, 2021, which is a period that was not included in model training. Consistent with the main model, we also performed nested tenfold cross-validation, described in detail in the Supplementary Material, and independently validated the model in the CRUSH Covid dataset.”

Updated paragraph in the Discussion, page 14:

“We observed a peak in app-based COVID-19 prevalence estimates in mid-September 2020 with no corresponding peak in any disease-specific COVID-19 national register data. The prevalence of loss of smell and/or taste, sore throat, and headache were similarly elevated in NOVUS. The Public Health Agency of Sweden also noted high occurrence of symptoms of acute respiratory infections at this time⁶. Laboratory analyses of respiratory viruses later indicated a high incidence of common colds caused by rhinoviruses in September 2020⁷. Hence, the specificity of the CSSS data was compromised when prevalence of other pathogens with similar symptomatology to COVID-19 was elevated. We therefore added time as a variable in the model developed in Step 1, allowing estimated probabilities to vary depending on the PCR test positivity rate during a given period. This second model yielded results more consistent with the national COVID-19 incidence data but is strongly influenced by the proportion of positive COVID-19 tests. The model is thus rather insensitive to a sudden increase or decrease in the reporting of symptoms, and change in prevalence will not be fully captured until this trend is also reflected in the test results reported in the app. The more static main model will, conversely, react more quickly to an increase of reported symptoms, raising the sensitivity of the model but also the risk of false positive healthcare alerts. Because of the delay inherent in this type of analysis, the time-dependent model is not suitable for real-time COVID-19 surveillance; it is also ineffective when test positivity varies greatly across regions, unless regions are modelled separately. An intermediate solution would be to retrain the model at known events that affect the relationship between symptoms and positive PCR tests, such as when vaccinations are introduced, new variants or other concurrent epidemics

emerge. Very few cases of seasonal influenza were confirmed in Sweden in the winter season of 2020/2021 compared with previous years⁸, which rendered the lower specificity during this period less problematic.”

Comment 2c: AUC scores of ~0.75 don't support the quality of the predictions made by the data and the model. The observed spike in September 2020, as also commented on in the Discussion, also doesn't support the prediction quality. So, the conclusions in the manuscript on the adoptability of the app should be toned down accordingly.

Authors' reply: The aim in the current study was not classification on the individual where a higher AUC would be wanted, but rather to estimate nationwide and regional prevalence of symptomatic COVID-19 and to predict next week hospital admissions. For this aim to be feasible, we needed the model to be able to 1) discriminate reasonably well between symptomatic COVID-19 and symptoms caused by other infections or diseases, as evaluated by the AUC, and 2) to generate reasonable COVID-19-probabilities (as evaluated by calibration curves).

With regards to the September peak, we agree that this highlights the main weakness of syndromic surveillance, when other infections compete with overlapping symptoms. We have shown that including time-varying coefficients might help to decrease the chance of these false warnings, however, such modelling comes at a price. We have now discussed this in more detail in the discussion.

Updated paragraphs in the Discussion, pages 11 and 14:

“Adequate and continuous regional COVID-19 surveillance is challenging and requires multiple sources of data. Here, we developed an app-based framework that allowed for syndromic surveillance of COVID-19 at the national and regional level in Sweden across the first two pandemic waves. We found that CSSS prevalence estimates could be used to monitor COVID-19 disease trends, and that they were particularly informative in times of limited PCR testing capacity. **The accuracy of the prevalence estimates was, however, lower in September 2020 when other respiratory infections peaked.** We further showed that combining app-based regional prevalence estimates with previously recorded hospital data could, with moderate accuracy, predict regional levels of COVID-19 hospital admissions the following week both in Sweden and in England.”

and

“We observed a peak in app-based COVID-19 prevalence estimates in mid-September 2020 with no corresponding peak in any disease-specific COVID-19 national register data. The prevalence of loss of smell and/or taste, sore throat, and headache were

similarly elevated in NOVUS. The Public Health Agency of Sweden also noted high occurrence of symptoms of acute respiratory infections at this time⁶. Laboratory analyses of respiratory viruses later indicated a high incidence of common colds caused by rhinoviruses in September 2020⁷. Hence, the specificity of the CSSS data was compromised when prevalence of other pathogens with similar symptomatology to COVID-19 was elevated. We therefore added time as a variable in the model developed in Step 1, allowing estimated probabilities to vary depending on the PCR test positivity rate during a given period. This second model yielded results more consistent with the national COVID-19 incidence data but is strongly influenced by the proportion of positive COVID-19 tests. The model is thus rather insensitive to a sudden increase or decrease in the reporting of symptoms, and change in prevalence will not be fully captured until this trend is also reflected in the test results reported in the app. The more static main model will, conversely, react more quickly to an increase of reported symptoms, raising the sensitivity of the model but also the risk of false positive healthcare alerts. Because of the delay inherent in this type of analysis, the time-dependent model is not suitable for real-time COVID-19 surveillance; it is also ineffective when test positivity varies greatly across regions, unless regions are modelled separately. An intermediate solution would be to retrain the model at known events that affect the relationship between symptoms and positive PCR tests, such as when vaccinations are introduced, new variants or other concurrent epidemics emerge. Very few cases of seasonal influenza were confirmed in Sweden in the winter season of 2020/2021 compared with previous years⁸, which rendered the lower specificity during this period less problematic.”

Comment 2d: The ground truth, not just the predictions, should be included in the plots in Figure 6 and other similar plots in the rest of the paper. This is important for assessing the accuracy of the trends observed.

Authors' reply: Thank you for this constructive comment. We have now added COVID-19 case notification rates from the Public Health Agency of Sweden and COVID-19 hospitalization rates obtained from the National Patient Registers to Figures 4, 5, and 6a (numbering updated to Figure 3a-d and Supplementary Figure 5a). We could not add such trend curves to 6b (updated to Supplementary Figure 5b), depicting estimated prevalence of COVID-19 in healthcare professionals and non-healthcare professionals, as data on case notifications rates and/or hospitalization rates are not available by occupation in Sweden. Further, as diagnostic PCR testing in Sweden was limited to healthcare workers and hospital inpatients until late June 2020⁹, case notification rates during the spring of 2020 do not reflect community transmission rates. We have also added the time point and prevalence of the national point prevalence survey performed by the Public Health Agency of Sweden in late May 2020.

Updated Figure 3, on page 38:

Figure 3. National prevalence estimate, with 95% confidence interval, of symptomatic COVID-19 in COVID Symptom Study Sweden (main model utilized for real-time prediction estimates, and retrospective time-dependent model), combined in a) and c) with retrospective data on daily number of new hospital admissions registered in the National Patient Register per 100,000 inhabitants ≥ 18 years, and in b) and d) with daily number of new COVID-19 cases registered in SmiNet, per 100,000 inhabitants ≥ 18 years.

* Time-point for recalibration of CSSS national COVID-19 prevalence estimate using national point prevalence survey findings from the Public Health Agency of Sweden

and

Updated Supplementary Figure 5, on page 11 in the Supplementary Material:

Supplementary Figure 5. National prevalence estimates of symptomatic COVID-19 in COVID Symptom Study Sweden, depicting the main model (utilized for real-time prediction estimates) and the retrospective time-dependent model, stratified by a) sex and age (18–39, 40–64, and ≥65 years), and b) sex and age (18–39, 40–64, and ≥65 years) and healthcare professional (HP). **Panel a) is further combined with retrospective data on daily number of new COVID-19 cases registered in SmiNet, per 100,000 inhabitants ≥18 years**

a)

b)

Addressing these and other reviewers' comments should strengthen the manuscript significantly.

Reviewer #2

This is a well-controlled study to assess whether a symptoms-based App can be useful to predict COVID-19 incidence and hospitalizations. Such information is useful for tracking infection during the pandemic and preparedness for health facilities. In most respects the analyses are well done and the conclusions solid. Limitations of the study are the novelty and minor comments below.

Authors' reply: We wish to thank the Reviewer for these encouraging remarks.

Comment 1: A similar set of studies has been published by the UK group using a much larger population (reference 8).

Authors' reply: Thank you. The article *Detecting COVID-19 infection hotspots in England using large-scale self-reported data from a mobile application: a prospective, observational study* by Varsavsky et al. (2021)¹⁰ reported that data from COVID Symptom Study England could be used to detect rapid increases in community transmission. However, the study period of that paper was March to September 2020, only encompassing the first pandemic wave in England. We validated and expanded their findings to also include a) data collected in a separate European country, b) the second pandemic wave, with data until February 7, 2021, and c) the potential of app-based syndromic surveillance to predict subsequent hospitalisations. In this revision, we have further validated the hospital admission prediction model in an English dataset without additional local test data. Real-time and accurate prediction of coming hospital admissions may provide public health decision makers, as well as healthcare systems, with the time and opportunity to redirect limited resources when necessary; such predictions are also of high public interest. We have now updated this section of the Discussion.

Updated sections in the Discussion, pages 11-12:

“A previous study from ZOE COVID Study demonstrated how app data from the first pandemic wave from March through September 2020 could be utilized to successfully identify emerging COVID-19 hotspots in England, with findings validated in UK Government test data¹⁰. The present study confirmed the utility of app-based COVID-19 syndromic surveillance, encompassing the full second pandemic wave in the Swedish population and expanding the scope to include predictions of subsequent

hospital admissions. The validation of the CSSS-based hospital prediction model in English data highlights the potential transferability of our approach, without using any PCR test data in the English data. Syndromic surveillance of COVID-19 may thus provide early warnings of surges in hospital admissions, thereby helping guide the allocation of precious healthcare resources in times of crisis.”

and

“The accuracy of the CSSS-based hospital prediction model during the first and second wave was higher for more populated regions in Sweden. Moreover, when the Swedish model was applied in England across healthcare regions, MdAPEs were lower than those derived in the Swedish setting. Although we cannot discern the separate influences of larger total population size, higher absolute number of study participants, and higher study participation rates, it is likely that all these factors enhance the accuracy of the hospital prediction model.

When we compared the CSSS-based hospital prediction model with the SMI-Net-based (PCR test-based) hospital prediction model, we observed that the accuracy of the CSSS-based model was higher during the first wave, while the SMI-Net-based model was similar in the second wave. The higher accuracy of CSSS during the first pandemic wave is likely due to the limited availability of PCR testing in Sweden at that time, when tests were only available to hospital inpatients and healthcare workers.⁹ We conclude that in addition to the expansion of the national PCR testing programme introduced in June 2020 and subsequent delays in PCR testing, local factors may also influence how well the CSSS app and PCR testing efforts reflect regional and/or local outbreaks. However, the successful application of the non-test dependent hospital prediction model in England shows great promise for future efforts in syndromic surveillance, where models can be trained in one country with dense test data and adjusted to local trends of hospitalizations in a second country without the need for additional test data.”

Comment 2: The authors could advance the field further by performing more in depth analysis about the effects of a) medication b) ethnicity c) economic status d) comorbidities on hospitalizations. This would add considerable novelty beyond what is already published.

Authors' reply: Thank you for this interesting suggestion. The updated regional hospitalization prediction model uses current COVID Symptom Study Sweden (CSSS) prevalence estimates and current hospital admissions as the independent variables, and new hospital admissions seven days ahead as the dependent variables. However, the data underlying these two variables stem from two separate data sources, with varying availability of the mentioned data.

The CSSS prevalence estimates of COVID-19 in the general population are based on, a) the daily individual probability of symptomatic COVID-19 in all CSSS study participants, assessed using our own symptom-based algorithm, and b) the weighted average of those individual probabilities applied to the general population in Sweden.

As for hospitalization data, we extracted an anonymized population-based dataset from national population and health registers held by the Swedish Board of Health and Welfare and Statistics Sweden. The dataset included all individuals hospitalized for COVID-19 from January 1, 2020 until January 4, 2021, as well as information on sex, year of birth, and home address postal code. However, we do not have access to information on ethnicity, pre-existing health conditions, medications, or individual-level socioeconomic variables in this data set, and we cannot link this anonymized dataset to study participants in COVID Symptom Study Sweden.

To address the Reviewer's concern about potential influence of socioeconomic circumstances on hospitalization predictions, we explored data from the entire study period in Swedish data to assess which variables to include in the model predicting hospital admission seven days ahead. The candidate variables were: daily regional CSSS prevalence estimate on day 0 or -1; current regional rate of COVID-19 hospitalizations per 100,000 inhabitants ≥ 18 years on day 0 or -1; weekday of hospitalization (Monday through Sunday); mean age in region; and mean regional Neighbourhood Deprivation Index (NDI). The final model was a weighted linear regression model, including daily regional CSSS prevalence estimate on day 0 and daily regional rate of COVID-19 hospitalizations per 100,000 inhabitants ≥ 18 years on day 0, assuming both variables to have a linear relationship with the outcome. Mean NDI and mean age did however not notably influence the predictions and were therefore not included. Although NDI and mean age were associated with the hospitalizations seven days ahead, they were omitted from the model as they did not notably improve the predictions, likely because of collinearity with current hospital admission.

Updated section on the Methods, on page 23:

“Step 4. Predictions of regional COVID-19 hospital admissions the following week

We assessed the ability of the CSSS prevalence estimates to predict the following week’s COVID-19 hospitalizations during the first and second pandemic waves in Sweden. We defined the end of the first pandemic wave as when the rate of daily new hospital admissions in Sweden dropped below 0.5 individuals per 100,000 inhabitants ≥ 18 years (July 3, 2020), and the beginning of the second wave as when the hospital admission rate again rose above that (October 19, 2020).

We utilized data from the entire study period in Swedish data to assess which variables to include in the model predicting hospital admission seven days ahead. The candidate variables were: daily regional CSSS prevalence estimate on day 0 or -1; current regional rate of COVID-19 hospitalizations per 100,000 inhabitants ≥ 18 years on day 0 or -1; weekday of hospitalization (Monday through Sunday); mean age in region; and mean regional Neighbourhood Deprivation Index (NDI). The final model was a weighted linear regression model, including daily regional CSSS prevalence estimate on day 0 and daily regional rate of COVID-19 hospitalizations per 100,000 inhabitants ≥ 18 years on day 0, assuming both variables to have a linear relationship with the outcome. Mean age and mean NDI did not notably influence the predictions and were therefore not included. To create weights for the linear regression we multiplied the number of inhabitants ≥ 18 years in each region with a density assigned to each day. The density was created using a Gaussian kernel function, with the highest density for the most recent observation. The standard deviation for the kernel function was set to be two days by trial-and-error. Weights for days more recent than seven days prior to the observation being forecasted was set to zero.”

Comment 3: Line 28 Selection of one PCR random test seems problematic. If multiple tests were performed in a short period and initial positive test immediately followed by several negative tests is likely a false positive. Similarly, a negative test follow by several positive test is likely a positive case at least at the downstream tests. I think this could be adjudicated better.

Authors' reply: Thank you for highlighting this issue. We had 19,161 CSSS study participants reporting at least one PCR test result included in the development of our model predicting the probability of a positive COVID-19 test result. Out of these 19,161 participants, 107 had reported >1 PCR test result with divergent results within +/- 10 days of the random PCR test included in the prediction model development. As a sensitivity analysis, we therefore excluded these 107 individuals from the model. The coefficients however remain largely unchanged (please find a table with coefficients from the main model and from the sensitivity analysis below). We further compared the main model national prevalence estimates with national prevalence estimates produced by the sensitivity analysis (please find figure included below). As we could not detect any differences in prevalence estimates, we have not included the sensitivity analyses in the main manuscript, but we are open to this if the Reviewer so prefers.

Table A (not included in manuscript). Coefficients from the main prediction model and the sensitivity analysis excluding 107 individuals with >1 PCR test result with divergent results within +/- 10 days of the random PCR test included in the prediction model development

	MAIN MODEL n tests=19,161	Sensitivity analysis n tests=19,054
Variable	Coefficient	Coefficient
Loss of smell and/or taste	2.42	2.47
Fever	1.04	1.07
Unusual muscle pains	0.71	0.70
Persistent cough	0.46	0.45
Male sex	0.40	0.42
Headache	0.26	0.28
Skipped meals	0.14	0.13
Fatigue	0.13	0.09
Chest pain	0.10	0.11
Dizzy, light-headedness	0.00	0.02

Hoarse voice	-0.03	-0.03
Delirium	-0.24	-0.23
Blisters on feet	-0.31	-0.33
Abdominal pain	-0.31	-0.32
Shortness of breath	-0.34	-0.36
Red welts on face or lips	-0.35	-0.39
Nausea	-0.37	-0.38
Sore throat	-0.55	-0.55
(Intercept)	-4.86	-4.86
Loss of smell and/or taste x Nausea	0.33	0.35
Loss of smell and/or taste x Diarrhoea	0.30	0.29
Loss of smell and/or taste x Dizzy, light-headedness	0.22	0.20
Loss of smell and/or taste x Abdominal pain	0.10	0.09
Loss of smell and/or taste x Red welts on face or lips	-0.06	-
Loss of smell and/or taste x Skipped meals	-	-0.01
Loss of smell and/or taste x Fatigue	-0.09	-0.02
Loss of smell and/or taste x Chest pain	-0.09	-0.12
Loss of smell and/or taste x Male sex	-0.14	-0.15
Loss of smell and/or taste x Headache	-0.16	-0.18
Loss of smell and/or taste x Hoarse voice	-0.28	-0.29
Loss of smell and/or taste x Eye soreness	-0.29	-0.28
Loss of smell and/or taste x Unusual muscle pains	-0.31	-0.29
Loss of smell and/or taste x Sore throat	-0.37	-0.37
Loss of smell and/or taste x Persistent cough	-0.44	-0.44
Loss of smell and/or taste x Fever	-0.75	-0.76

Figure A (not included in manuscript). National prevalence estimates, with 95% confidence interval, of symptomatic COVID-19 in COVID Symptom Study Sweden. Main model is based on 19,161 PCR test results from study participants. The sensitivity analyses exclude the 107 individuals who had reported >1 PCR test result and had diverging test results within +/- 10 days of the random PCR test included in the main model development (n tests=19,054).

Comment 4: Did the authors try other modelling methods besides LASSO?

Authors' reply: Thank you. We found LASSO suitable for this study since our aim was to generate probabilities, something that machine learning methods, such as random forests or support vector machines (while excellent for binary classification), are less suited for. We did not want to use ridge penalties as we needed some variable selection for our model. Also, a fair evaluation of model discrimination (i.e. AUC) would have been trickier if we included other models in the process.

We did, however, consider a sensitivity analysis using smoothly clipped absolute deviations (SCAD)¹¹. This would shrink the most important variables less than LASSO, and the less important variables more. We have now performed such an analysis and include the SCAD coefficients as well as the national prevalence estimates comparing Main model (LASSO) with Main model (SCAD) below (the curves are overlapping and are difficult to distinguish). As we could not detect any advantages to using SCAD, we have not included this sensitivity analysis in our manuscript. We are, however, happy to include it if the Reviewer so prefers. The SCAD(AUC) is 0.77 (95% CI 0.76–0.78) for the training data (April 29 to December 31, 2020), 0.72 (95% CI 0.69–0.76) for the test data (January 1 to February 10, 2021) and 0.79 (95% CI 0.74–0.83) for validation in the external dataset CRUSH Covid.

Table B (not included in manuscript). Coefficients for Main model (LASSO) and Main model (SCAD).

	Main model LASSO	Main model SCAD
Variable	Coefficient	Coefficient
Loss of smell and/or taste	2.42	2.60
Fever	1.04	1.10
Unusual muscle pains	0.71	0.74
Persistent cough	0.46	0.48
Male sex	0.40	0.45
Headache	0.26	0.33
Skipped meals	0.14	0.15
Fatigue	0.13	0.08
Chest pain	0.10	0.16
Dizzy, light-headedness	0.00	-
Hoarse voice	-0.03	-

Delirium	-0.24	-0.26
Blisters on feet	-0.31	-0.29
Abdominal pain	-0.31	-0.38
Shortness of breath	-0.34	-0.41
Red welts on face or lips	-0.35	-0.43
Nausea	-0.37	-0.44
Sore throat	-0.55	-0.56
(Intercept)	-4.86	-4.92
Loss of smell and/or taste x Nausea	0.33	0.43
Loss of smell and/or taste x Diarrhoea	0.30	0.32
Loss of smell and/or taste x Dizzy, light-headedness	0.22	0.25
Loss of smell and/or taste x Abdominal pain	0.10	0.19
Loss of smell and/or taste x Red welts on face or lips	-0.06	-
Loss of smell and/or taste x Fatigue	-0.09	-
Loss of smell and/or taste x Skipped meals	-	-0.01
Loss of smell and/or taste x Chest pain	-0.09	-0.20
Loss of smell and/or taste x Male sex	-0.14	-0.22
Loss of smell and/or taste x Headache	-0.16	-0.26
Loss of smell and/or taste x Hoarse voice	-0.28	-0.34
Loss of smell and/or taste x Eye soreness	-0.29	-0.31
Loss of smell and/or taste x Unusual muscle pains	-0.31	-0.35
Loss of smell and/or taste x Sore throat	-0.37	-0.39
Loss of smell and/or taste x Persistent cough	-0.44	-0.49
Loss of smell and/or taste x Fever	-0.75	-0.82

Figure B (not included in manuscript). National prevalence estimates, with 95% confidence interval, of symptomatic COVID-19 in COVID Symptom Study Sweden, depicting Main model (LASSO) and Main model (SCAD).

Reviewer #3

The manuscript presents an impressive COVID surveillance effort in Sweden based on self-reported information on a digital app.

The researchers develop a model which can reasonably differentiate between users reporting positive vs negative PCR test results based on a range symptoms and some demographic data. The model is developed using cross-validation methods, tested on a period not included in the training data, and then tested again on external data from the CRUSH survey in Sweden.

The researchers then use the model to estimate and predict prevalence of COVID cases and hospitalizations across space and time in Sweden. After initial calibration to the FoHM survey and adjustment to differences in demographics, the researchers report various metrics showing correlation and agreement between the model-based estimates and reported cases and hospitalizations.

The work is impressive in scope. The methods are overall very transparent and clear regarding the methodological choices, and the study employs several internal and external methods to validate the results.

Authors' reply: We would like to thank the Reviewer for these kind words of support and encouragement.

Comment 1: The description regarding the inclusion/exclusion of participants reporting symptoms and PCR tests within a week of joining was unclear and seemed contradictory.

Authors' reply: Thank you. The rationale for not including symptoms from CSSS participants within the first week after joining was that we wanted to avoid the potential inflation of prevalence estimates that could occur if participants were more likely to join the study and report in the app when they were experiencing potential COVID-19 symptoms than if they were healthy. To further illustrate this, we have now assessed the proportion of all study participants who reported any symptoms within seven days of joining the study, and after the first seven days, respectively. We found that across the entire study period (April 29, 2020 to February 10, 2021), a higher proportion of study participants reported symptoms within the first week after joining than thereafter. We have now clarified this rationale in the manuscript, and we have also included the new figure below in the Supplementary Material.

Updated section on the Methods, on page 19:

“Step 1. Training of model for estimation of individual probability of COVID-19

We trained a model to estimate the individual probability of a symptomatic COVID-19, defined as having symptoms and a positive PCR test result. The model training set consisted of data from 19,161 participants who reported at least one PCR test result (of which 2,588 were positive) between April 29 and December 31, 2020, and who reported at least one reported candidate symptom within seven days before or on the test date. As we observed that a higher proportion of study participants reported symptoms within the first week of joining the study than thereafter (Supplementary Figure 8), we excluded all reports submitted during the first week to reduce participation bias from increased motivation among symptomatic individuals in the general population. For participants who had not submitted reports every day, we assumed the last reported observation to be valid for no more than seven subsequent days. If a participant submitted more than one report on a given day, all reports were combined into a single daily report; a symptom was treated as reported if it was mentioned in at least one of these reports.”

New supplementary figure in the Supplementary Material, page 16:

“Supplementary Figure 8. Proportion of CSSS study participants reporting any symptom within seven days of joining and thereafter, respectively.

Comment 2: The concepts proposed by the study, whereby self reported symptoms can be used to identify disease in the individual, and to help support surveillance and monitoring of infectious diseases in the population are not original, and have been explored by dozens of studies both prior to COVID and during COVID.

For instance this systematic review summarizes over 169 studies suggesting models to predict disease and severity in the individual (<https://www.bmj.com/content/369/bmj.m1328.long>), this systematic review surveys over 750 studies on use of digital technologies for disease surveillance (<https://www.nature.com/articles/s41746-021-00407-6>), and these are notable examples for COVID (<https://www.nature.com/articles/s41591-020-0916-2>, <https://www.science.org/doi/full/10.1126/science.abc0473>, <https://www.nature.com/articles/s41591-020-0857-9>).

Authors' reply: Thank you for this constructive comment. We agree that we had not contextualised our study sufficiently within the field of participatory syndromic surveillance. We have therefore updated our introduction to better introduce other relevant and important early eHealth COVID-19 efforts. The aim in the introduction has also been updated to better reflect the novelty of our study.

Updated Introduction, on pages 4-5:

“INTRODUCTION

Real-time and accurate COVID-19 disease surveillance data is critical for adequate public health decision making and evaluation, as well as for healthcare system preparedness. The WHO guidelines for COVID-19 surveillance highlight the importance of combining data from multiple surveillance systems, and how participatory syndromic surveillance, where participants self-report symptoms of possible infection, may constitute a useful tool in early detection of disease outbreaks¹². The European Centre for Disease Prevention and Control further notes that the utility of COVID-19 participatory syndromic surveillance may be enhanced if symptom data can be combined with information on testing¹³. Expanding knowledge on the feasibility of large-scale syndromic surveillance may thus enable tailored population-based participatory surveillance initiatives in future pandemics and epidemics.

Several novel eHealth solutions aimed at real-time monitoring and prediction of the dynamics of COVID-19 transmission were introduced early in the pandemic^{14, 15, 16, 17}, with app-based technologies quickly recognized as a potentially powerful surveillance

tool. One of these technologies was the ZOE COVID Symptom Study app, designed to collect baseline health data as well as daily reports on symptoms and test results from study participants. The app was launched in the United Kingdom and in the United States in late March 2020^{18, 19, 20}.

Community transmission of SARS-CoV-2 was confirmed in Sweden in early March 2020. However, during the first pandemic wave in the spring of 2020, PCR testing was only available for hospital inpatients and healthcare workers⁹ in Sweden and assessments of national COVID-19 prevalence were based on two PCR surveys performed by the Public Health Agency of Sweden in April (n=2,571) and May (n=2,957)²¹. Nationwide PCR testing for symptomatic adults was later introduced in June 2020⁹, but suffered from various issues such as long analysis times during periods of high demand²². In response to the limited surveillance during the first pandemic wave, the COVID Symptom Study was launched in Sweden in April 2020.

The aims of this study were to develop and evaluate a syndromic surveillance-based framework to estimate the regional prevalence of COVID-19 and to evaluate if these could be used to accurately predict subsequent trends in COVID-19 hospital admissions. We showed that a model trained on symptoms and test data could provide informative prevalence estimates, and contribute to predictions of hospital burden the following week. Without using any additional test data, the hospital prediction model further performed well outside Sweden in a second country, England.”

Comment 3: The model is shown to be predictive on retrospective Swedish data in a limited time-frame. As the model is clearly tailored to the COVID experience and monitoring systems in Sweden, I don't believe the authors provide evidence or rationale which would lead one to expect the model would be generalizable outside of Sweden. It is doubtful performance would remain consistent in Sweden itself. Indeed the authors note the model displayed a non-existent COVID "wave", and therefore provide a more elaborate time-dependent model, which can only be fit to the data retroactively.

Authors' reply: Thank you. We fully agree that demonstrating generalizability of our results to another country would add significant value to our findings. We have therefore developed the hospital prediction model to make it more generalizable by including current regional hospital admissions in each prediction. We have also performed a validation of our hospital prediction model in an external English dataset, showing rather encouraging results. This dataset included daily reports from 2,638,536 COVID Symptom Study England study participants as well as information on 318,232 COVID-19 hospital admissions (n=318,232) from April 6, 2020 to February 7, 2021 extracted from National Health Service England. No local data on testing was however included when validating the findings to simulate a situation with low testing – neither from COVID Symptom Study England nor from national English registers.

We applied the exact same algorithm that was developed in the first step in CSSS to estimate daily individual probability of COVID-19 in the English dataset, and then estimated the daily age- and sex-weighted COVID-19 prevalence across the English regions. We further applied the same iterative time-updated prediction model as in the Swedish data set to predict hospital admissions the following week in the seven English regions. We used available outcome data up to April 27, 2020 to tune the parameters and to predict admissions on May 4, 2020 and then we repeated this daily throughout the study period (until February 7, 2021). Across the seven English healthcare regions, we observed MdAPEs of 22.3% and 19.0% (first and second pandemic wave, respectively).

In conclusion, a prediction model of COVID-19 hospital admissions, based on regional prevalence estimates from syndromic surveillance in combination with current hospital data in Sweden, may predict hospital admissions without local test data in the second country We have now updated the manuscript to include the updates of the hospital prediction model and this external validation.

Updated Title, on page 1:

**“App-based COVID-19 syndromic surveillance and prediction of hospital admissions:
The COVID Symptom Study Sweden”**

Updated Abstract, on page 3:

“ABSTRACT

The app-based COVID Symptom Study was launched in Sweden in April 2020 to contribute to real-time COVID-19 surveillance. We enrolled 143,531 study participants (≥18 years) who contributed 10.6 million daily symptom reports between April 29, 2020 and February 10, 2021. Data from 19,161 self-reported PCR tests were used to create a symptom-based model to estimate the individual probability of symptomatic COVID-19, with an AUC of 0.78 (95% CI 0.74–0.83) in an external dataset. These individual probabilities were used to estimate daily regional COVID-19 prevalence, which were in turn used together with current hospital data to predict next week COVID-19 hospital admissions. We found that this hospital prediction model demonstrated a lower median absolute percentage error (MdAPE: 25.9%) across the five most populated regions in Sweden during the first pandemic wave than a model based on case notifications (MdAPE: 30.3%). During the second wave, the error rates were similar. When applying the same model to an English dataset, not including local COVID-19 test data, we observed MdAPEs of 22.3% and 19.0%, respectively, highlighting the transferability of the prediction model.”

Updated aims in the Introduction, on page 5:

“The aims of this study were to develop and evaluate a syndromic surveillance-based framework to estimate the regional prevalence of COVID-19 and to evaluate if these could be used to accurately predict subsequent trends in COVID-19 hospital admissions. We showed that a model trained on symptoms and test data could provide informative prevalence estimates, and contribute to predictions of hospital burden the following week. Without using any additional test data, the hospital prediction model further performed well outside Sweden in a second country, England.”

New paragraph in the Results, on pages 9-10:

“Step 5. Validation of the hospitalization prediction model in England

We sought to validate the CSSS-based hospitalization prediction model in England by repeating Steps 2 and 3 and parts of Step 4. The English dataset encompassed daily reports from 2,638,536 ZOE COVID Study (CSS UK) English study participants from March 30, 2020 to January 31, 2021 (study population characteristics and regional participation rates are available in Supplementary Tables 3 and 4). We extracted information on all COVID-19 hospital admissions (n=318,232) in individuals ≥ 18 years across the seven English healthcare regions from April 6, 2020 through February 7, 2021 from National Health Service England data. The number of new daily COVID-19 hospital admissions ranged from 0–958 across the English regions during this period.

We applied the exact same model that was developed in Step 1 in CSSS to estimate daily individual probability of COVID-19 in the English dataset, and then estimated the daily age- and sex-weighted COVID-19 prevalence across the English regions. We further applied the same iterative time-updated prediction model as in the Swedish dataset to predict hospital admissions the following week in the seven English regions. We used available outcome data up to April 27, 2020 to tune the parameters and to predict admissions a week later on May 4, 2020 and then repeated this daily throughout the study period (until February 7, 2021).

Across the seven English regions, we observed an MdAPE of 22.3% for the part of the first English pandemic wave captured in the data (April 6 to June 19, 2020) and an MdAPE of 19.0% for the second English wave (September 20, 2020 to February 7, 2021; Figure 5, Supplementary Table 5). As in Sweden, we observe lower error in the most populated English healthcare region (West and East Midlands; population ≥ 18 years n=10.8 million) with MdAPE of 16.1% and 14.0%, respectively. Overall, the predicted number of hospital admissions were overestimated when daily regional hospital admissions were low (Supplementary Figure 7b).”

New paragraphs in the Discussions, on pages 11-12:

Adequate and continuous regional COVID-19 surveillance is challenging and requires multiple sources of data. Here, we developed an app-based framework that allowed for syndromic surveillance of COVID-19 at the national and regional level in Sweden

across the first two pandemic waves. We found that CSSS prevalence estimates could be used to monitor COVID-19 disease trends, and that they were particularly informative in times of limited PCR testing capacity. The accuracy of the prevalence estimates was, however, lower in September 2020 when other respiratory infections peaked. We further showed that combining app-based regional prevalence estimates with previously recorded hospital data could, with moderate accuracy, predict regional levels of COVID-19 hospital admissions the following week both in Sweden and in England.

A previous study from ZOE COVID Study demonstrated how app data from the first pandemic wave from March through September 2020 could be utilized to successfully identify emerging COVID-19 hotspots in England, with findings validated in UK Government test data¹⁰. The present study confirmed the utility of app-based COVID-19 syndromic surveillance, encompassing the full second pandemic wave in the Swedish population and expanding the scope to include predictions of subsequent hospital admissions. The validation of the CSSS-based hospital prediction model in English data highlights the potential transferability of our approach, without using any PCR test data in the English data. Syndromic surveillance of COVID-19 may thus provide early warnings of surges in hospital admissions, thereby helping guide the allocation of precious healthcare resources in times of crisis.”

and

“The accuracy of the CSSS-based hospital prediction model during the first and second wave was higher for more populated regions in Sweden. Moreover, when the Swedish model was applied in England across healthcare regions, MdAPEs were lower than those derived in the Swedish setting. Although we cannot discern the separate influences of larger total population size, higher absolute number of study participants, and higher study participation rates, it is likely that all these factors enhance the accuracy of the hospital prediction model.

When we compared the CSSS-based hospital prediction model with the SMI-net-based (PCR test-based) hospital prediction model, we observed that the accuracy of the CSSS-based model was higher during the first wave, while the SMI-net-based model was similar in the second wave. The higher accuracy of CSSS during the first pandemic wave is likely due to the limited availability of PCR testing in Sweden at that time, when tests were only available to hospital inpatients and healthcare workers.⁹ We

conclude that in addition to the expansion of the national PCR testing programme introduced in June 2020 and subsequent delays in PCR testing, local factors may also influence how well the CSSS app and PCR testing efforts reflect regional and/or local outbreaks. However, the successful application of the non-test dependent hospital prediction model in England shows great promise for future efforts in syndromic surveillance, where models can be trained in one country with dense test data and adjusted to local trends of hospitalizations in a second country without the need for additional test data.”

New section in the Methods, on pages 25-26:

“Step 5. Validation of hospital prediction model using external data from England by repeating Steps 2–4

We repeated Step 2 in English data by applying the Swedish model for estimation of individual probabilities from Step 1, applying the same symptom coefficients as in Sweden, and thereby assessed the daily individual probability of symptomatic COVID-19 in ZOE COVID Study (CSS UK) English participants ≥ 18 years from March 30, 2020 to January 31, 2021. We included participants residing in all postal code areas ($n=2,261$) within any of the seven English healthcare regions (South East, London, North West, East of England, South West, West and East Midlands, Yorkshire and the Humber and North East; total population ≥ 18 years $n=43.2$ million). If postal code areas overlapped two or more regions the postal code area was randomly assigned to one of them in our analyses.

We further repeated Step 3 and assessed daily age- and sex-weighted averages of the individual probabilities to estimate daily COVID-19 prevalence across the seven English healthcare regions. Demographic data was extracted from the UK Office for National Statistics³.

We concluded by seeking replication of Step 4 by applying Eq. 1 in the English dataset using the equivalent variables used in the analysis of CSSS data. We trained the model in available outcome data up to the first day 0 (April 27, 2020) to derive the coefficients β_0 , $\beta_{\text{prevalence}}$, and β_{hospital} . We inserted the ZOE COVID Study prevalence estimate and daily regional rate of COVID-19 hospitalizations per 100,000 inhabitants ≥ 18 years on April 27, 2020 into the equation, applying the derived coefficients to predict hospitalization rates on May 4, 2020. We then repeated the model fit and

prediction from May 5, 2020 to February 11, 2021, with past data influencing the daily new intercept as well as the daily new two betas.

Information on COVID-19 hospital admissions from April 6, 2020 to February 7, 2021, was obtained from the UK government COVID-19 dashboard⁴. We defined the end of the first wave (June 19, 2020), and the start of the second (September 20, 2020), using the same threshold as in Swedish data. We calculated the MdAPEs between predicted and observed number of hospitalizations for each of the seven English healthcare regions during the first wave (April 6 to June 19, 2020), the summer period (June 20 to September 19, 2020), and the second wave (September 20, 2020 to February 7, 2021).”

New Figure on MdAPEs across English healthcare regions, on page 40:

Figure 5. Predicted number of daily hospital admissions seven days ahead across the seven English healthcare regions. The median absolute percentage errors (MdAPEs) of the predictions are denoted for the first pandemic wave (April 6 to June 19, 2020), the summer period (June 20 to September 19, 2020), and the second pandemic wave (September 20, 2020 to February 7, 2021).

New Supplementary Material, on pages 19-22 and 15:

Supplementary Table 3. Study population characteristics in COVID Symptom Study England.

	All	Women	Men
N (%) ¹	1,888,416 (100)	1,239,199 (65.6)	913,823 (34.4)
Age, years ²	47 (35, 59)	46 (35, 57)	50 (38, 61)
	≥65 (%)	155,573 (12.6)	121,545 (18.7)
Pregnant (%)	-	13394 (1.1)	-
BMI, kg/m ² ²	26 (23, 29)	25 (22, 29)	26 (24, 29)
	Obese, BMI ≥30 kg/m ² (%)	281,647 (22.7)	133,532 (20.6)
Current smoker (%)	84,036 (8.5)	53,386 (8.5)	30,650 (8.4)
Cardiovascular disease (%)	45,920 (2.4)	18,663 (1.5)	27,257 (4.2)
Antihypertensive medication (%)	160,698 (8.7)	81,885 (6.8)	78,813 (12.4)
Kidney disease (%)	12,961 (0.7)	8,027 (0.6)	4,934 (0.8)
Diabetes mellitus (%)			
	Yes, type 1	5,032 (0.4)	3,829 (0.6)
	Yes, type 2	18,140 (1.5)	22,164 (3.4)
	Yes, gestational	167 (<1)	4 (<1)
	Yes, other	683 (0.1)	453 (0.1)
	Yes, type not specified	5,103 (0.4)	5,593 (0.9)
Lung disease (%)			
	Yes, asthma only	133,602 (10.8)	57,215 (8.8)
	Yes, both asthma and lung disease	6,729 (0.5)	3,149 (0.5)
	Yes, lung disease only	8,475 (0.7)	5,893 (0.9)
	Yes, type not specified	28,929 (2.3)	13,077 (2.0)
Current cancer (%)	13,865 (1.4)	6,722 (1.1)	7,143 (2.0)
Immunosuppressive medication ³ (%)	63,669 (3.4)	43,298 (3.5)	20,371 (3.1)

Healthcare professional (%)				
	Interacts with patients	37,966 (4.1)	31,579 (5.5)	6,387 (1.8)
	Does not interact with patients	25,702 (2.8)	20,509 (3.5)	5,193 (1.5)
Months entering the study (%)				
	March 2020	101,4687 (53.7)	692,095 (55.9)	322,592 (49.7)
	April-May 2020	568,030 (30.1)	349,889 (28.2)	218,141 (33.6)
	June-July 2020	110,298 (5.8)	73,377 (5.9)	36,921 (5.7)
	August-September 2020	86,506 (4.6)	57,134 (4.6)	29,372 (4.5)
	October-November 2020	61,011 (3.2)	37,922 (3.1)	23,089 (3.6)
	December 2020-January 2021	41,540 (2.2)	24,981 (2.0)	16,559 (2.6)
	February 2021	6,344 (0.3)	3,801 (0.3)	2,543 (0.4)
Duration of study participation, days ^{4*}		182 (62, 305)	184 (64, 305)	178 (60, 305)

¹Row percentage, ²Median (first and third quartile), ³Corticosteroids, methotrexate and/or biological agents (treatment of cancer and/or rheumatic disease), ⁴From first to last daily report. BMI: Body Mass Index

and

Supplementary Table 4. COVID Symptom Study England participation rate across the seven English healthcare regions.

English healthcare region	Inhabitants ≥18 years	CSSS participants	CSSS participation rate per 100,000 inhabitants ≥18 years
East of England	4,787,130	232,430	4,855
London	6,775,147	328,087	4,843
Midlands	8,259,089	267,110	3,234
North East and Yorkshire	6,308,756	204,245	3,237
North West	5,632,066	181,538	3,223
South East	7,034,905	437,217	6,215
South West	4,422,690	238,919	5,402

and

Supplementary Table 5. COVID Symptom Study England (CSSS England) median absolute percentage errors (MdAPEs) for prediction of new daily hospitalizations, across the first pandemic wave (April 6 to June 19, 2020), the summer period (June 20 to September 19, 2020), and the second pandemic wave (September 20, 2020 to February 7, 2021). The prediction model used current regional CSS England prevalence estimates and hospital data to predict hospital admissions seven days ahead.

	Median absolute percentage errors (%)		
	First wave April 6 to June 19, 2020	Summer period June 20 to September 19, 2020	Second Wave September 20, 2020, to February 7, 2021
All seven regions combined	22.3	36.0	19.0
East of England	24.6	40.6	20.8
London	38.8	25.9	20.5
Midlands	16.1	32.5	14.0
North East and Yorkshire	13.8	26.0	14.4
North West	18.7	35.8	17.5
South East	21.4	33.7	24.5
South West	33.0	79.4	22.1

and

Supplementary Figure 7. Relative error in % per day and region in hospital admission predictions over the range of observed admissions, in a) Sweden (May 11 to November 29, 2020), and b) England (April 27, 2020 to February 7, 2021).

b)

REFERENCES

1. Kennedy B, *et al.* App-based COVID-19 syndromic surveillance and prediction of hospital admissions The COVID Symptom Study Sweden. <https://doi.org/10.5281/zenodo.6069218>.
2. Harrell Jr FE. *Regression modeling strategies: with applications to linear models, logistic and ordinal regression, and survival analysis*. Springer (2015).
3. UK Office for National Statistics. Office for National Statistics. <https://www.ons.gov.uk/>.
4. UK government. The official UK government website for data and insights on coronavirus (COVID-19). <https://coronavirus.data.gov.uk/>.
5. Oppedal K, Eftestol T, Engan K, Beyer MK, Aarsland D. Classifying dementia using local binary patterns from different regions in magnetic resonance images. *Int J Biomed Imaging* **2015**, 572567 (2015).
6. Public Health Agency of Sweden. Weekly report about COVID-19, week 39 (Veckorapport om covid-19, vecka 39). <https://www.folkhalsomyndigheten.se/globalassets/statistik-uppfoljning/smittsamma-sjukdomar/veckorapporter-covid-19/2020/covid-19-veckorapport-vecka-39-final.pdf> Accessed: March 1, 2021.
7. Karolinska Universitetslaboratoriet. Respiratory pathogens (Luftvägspatogener). <https://www.karolinska.se/globalassets/global/2-funktioner/funktion-kul/klinisk-mikrobiologi/epidemiologi/rapport-influensa--och-rs-virus-och-andra-luftvagspatogener.pdf> Accessed: March 1, 2021.
8. Public Health Agency of Sweden. The influenza season 2020-2021 (Influensasäsongen 2020–2021). https://www.folkhalsomyndigheten.se/globalassets/statistik-uppfoljning/smittsamma-sjukdomar/veckorapporter-influensa/2020-2021/influensasasongen-2020-2021-sasongssammanfattning-final-v2_23juli.pdf Accessed: July 30, 2021.
9. Ludvigsson JF. The first eight months of Sweden's COVID-19 strategy and the key actions and actors that were involved. *Acta Paediatr* **109**, 2459-2471 (2020).
10. Varsavsky T, *et al.* Detecting COVID-19 infection hotspots in England using large-scale self-reported data from a mobile application: a prospective, observational study. *Lancet Public Health* **6**, e21-e29 (2021).
11. Fan J, Li R. Variable Selection via Nonconcave Penalized Likelihood and its Oracle Properties. *Journal of the American Statistical Association* **96**, 1348-1360 (2001).
12. World Health Organization. Public health surveillance for COVID-19: Interim Guidance. 16 December 2020. Preprint at

<https://apps.who.int/iris/bitstream/handle/10665/337897/WHO-2019-nCoV-SurveillanceGuidance-2020.8-eng.pdf> (2020).

13. European Centre for Disease Prevention and Control. COVID-19 surveillance guidance. October 2021. Preprint at <https://www.ecdc.europa.eu/sites/default/files/documents/COVID-19-surveillance-guidance.pdf> (2021).
14. Denis F, *et al.* Epidemiological Observations on the Association Between Anosmia and COVID-19 Infection: Analysis of Data From a Self-Assessment Web Application. *J Med Internet Res* **22**, e19855 (2020).
15. Timmers T, Janssen L, Stohr J, Murk JL, Berrevoets MAH. Using eHealth to Support COVID-19 Education, Self-Assessment, and Symptom Monitoring in the Netherlands: Observational Study. *JMIR Mhealth Uhealth* **8**, e19822 (2020).
16. Yoneoka D, *et al.* Early SNS-Based Monitoring System for the COVID-19 Outbreak in Japan: A Population-Level Observational Study. *J Epidemiol* **30**, 362-370 (2020).
17. Rossman H, *et al.* A framework for identifying regional outbreak and spread of COVID-19 from one-minute population-wide surveys. *Nat Med* **26**, 634-638 (2020).
18. Chan AT, *et al.* The COronavirus Pandemic Epidemiology (COPE) Consortium: A Call to Action. *Cancer Epidemiol Biomarkers Prev* **29**, 1283-1289 (2020).
19. Drew DA, *et al.* Rapid implementation of mobile technology for real-time epidemiology of COVID-19. *Science* **368**, 1362-1367 (2020).
20. Menni C, *et al.* Real-time tracking of self-reported symptoms to predict potential COVID-19. *Nat Med* **26**, 1037-1040 (2020).
21. Public Health Agency of Sweden. The prevalence of COVID-19 in Sweden April 21-24 and May 25-28, 2020 (Förekomsten av covid-19 i Sverige 21–24 april och 25–28 maj 2020). https://www.folkhalsomyndigheten.se/contentassets/fb47e03453554372ba75ca3d3a6ba1e7/forekomstren-covid-19-sverige-21-24-april-25-28-maj-2020_2.pdf Accessed: April 29, 2021.
22. Almgren M, Björk J. Mapping of differences in the regions' efforts for sampling and contact tracing during the COVID-19 pandemic (Kartläggning av skillnader i regionernas insatser för provtagning och smittspårning under coronapandemin). <https://coronakommissionen.com/wp-content/uploads/2021/10/underlagsrapport-m-almgren-kartlaggning-av-skillnader-i-regionernas-insatser-for-provtagning-och-smittsparning-under-coronapandemin.pdf> Accessed: November 19, 2021.

Reviewers' Comments:

Reviewer #1:

Remarks to the Author:

I am happy to see the substantial improvements the authors have made to the paper to address the reviewers' comments, including mine. I recommend acceptance, as long as the paper addresses the following few minor comments:

1. Wherever prediction scores (AUC, AUPRC etc) are mentioned, the dataset and/or evaluation setup, e.g., cross-validation, XYZ test set etc., should be clearly mentioned. This is done for several of the external test sets, but not so clearly for subsets of the CSSS dataset itself.
2. It is explained in the response to comment 1b that the work is indeed conducting classification/discrimination, for which they are providing performance estimates in the form of AUC scores. AUPRC should be provided at all the places AUC is mentioned. At the very least, AUPRC should be provided at places where conclusions are being drawn based on the AUC scores.
3. It is still unclear if any cleaning/preprocessing was done before the data were analyzed. This must be discussed clearly to ensure transparency and replicability.
4. A completed TRIPOD checklist wasn't found in the submission. This should be provided.

Reviewer #2:

Remarks to the Author:

I am fine with the revised manuscript

Reviewer #3:

Remarks to the Author:

I commend the authors for the very thorough revision. The revised manuscript addresses the comments appropriately, and provides additional explanation, details and transparency regarding methods and results.

Point-by-point response to reviewer

RE: App-based COVID-19 syndromic surveillance and prediction of hospital admissions in COVID Symptom Study Sweden (NCOMMS-21-33106). Kennedy et al.

REVIEWERS' COMMENTS

Reviewer #1

I am happy to see the substantial improvements the authors have made to the paper to address the reviewers' comments, including mine. I recommend acceptance, as long as the paper addresses the following few minor comments:

Comment 1: Wherever prediction scores (AUC, AUPRC etc) are mentioned, the dataset and/or evaluation setup, e.g., cross-validation, XYZ test set etc., should be clearly mentioned. This is done for several of the external test sets, but not so clearly for subsets of the CSSS dataset itself.

and

Comment 2: It is explained in the response to comment 1b that the work is indeed conducting classification/discrimination, for which they are providing performance estimates in the form of AUC scores. AUPRC should be provided at all the places AUC is mentioned. At the very least, AUPRC should be provided at places where conclusions are being drawn based on the AUC scores.

Authors' reply: Thank you, we have now added this information in the manuscript.

Updated section of the Results, on page 7:

“Step 1. Training of model for estimation of individual probability of COVID-19

Our analysis strategy consisted of five steps, as illustrated in Figure 2. In the first step, we developed a model to estimate individual probability of symptomatic COVID-19, utilizing information from 19,161 CSSS participants who reported at least one PCR test (of whom 2,586 were COVID-19 infection positive) between April 29 and December 31, 2020; these individuals also reported at least one candidate symptom within seven days before or on the test date. The final model selected by LASSO included 17 symptoms and sex, as well as 2-way interactions between loss of smell and/or taste and 14 symptoms, and a 2-way interaction between loss of smell and/or taste and sex.

The ROC area under the curve (AUC), produced by nested tenfold cross-validation, for this main model was 0.76 (95% CI 0.75–0.78; PR(AUC) with 95% CI: 0.38, 95% CI 0.35–

0.40) during the training period (April 29 to December 31, 2020; n=19,161) and 0.72 (95% CI 0.69–0.75; PR(AUC) 0.40, 95% CI 0.35–0.45) during the evaluation period (January 1 to February 10, 2021; n=1,753). In an external dataset of 943 symptomatic individuals from the CRUSH Covid survey (144 positive; test positivity 15.3%; October 18, 2020 to February 10, 2021), the AUC, also from nested tenfold cross-validation, was 0.78 (95% CI 0.74–0.83; PR(AUC): 0.48, 95% CI 0.40–0.56). Calibration graphs are available in Supplementary Figure 3.”

Updated paragraph in the Supplementary Material, page 8:

“The ROC area under the curve (AUC), produced by nested tenfold cross-validation, for the time-dependent model was 0.84 (95% CI 0.83–0.85; PR(AUC) 0.49, 95% CI 0.46–0.50) for the training period (April 29 and December 31, 2020), and 0.72 (95% CI 0.69–0.75; PR(AUC): 0.38, 95% CI 0.33–0.44) during the evaluation period (January 1 to February 10, 2021). In an external dataset of 943 symptomatic individuals from the CRUSH Covid survey (144 positive; test positivity 15.3%; October 18, 2020 to February 10, 2021), the AUC, also from nested tenfold cross-validation, of the time-dependent model was 0.75 (95% CI 0.70–0.79; PC(AUC): 0.35, 95% CI 0.27–0.41).”

Updated paragraph in the Methods, page 20:

“We further assessed calibration and discrimination in CSSS data (1,753 participants, of which 339 tested positive) from January 1 to February 10, 2021, the period not included in model training. The nested tenfold cross-validation is described in detail in the Supplementary Material. Discrimination was quantified using the ROC area under the sensitivity-specificity curve (AUC) and the area under the precision-recall curve PR(AUC). The confidence interval for AUC was calculated using the DeLong method¹ and the confidence interval for PR(AUC) by using bootstrap. Model calibration was assessed by plots with estimated probabilities divided into deciles.”

Comment 3: It is still unclear if any cleaning/preprocessing was done before the data were analyzed. This must be discussed clearly to ensure transparency and replicability.

Authors' reply: Thank you for highlighting this issue. We have now provided the pre-processing code "2_preprocess_abbrev.R" at <https://github.com/ulpha881/App-based-COVID-19-syndromic-surveillance-and-prediction-of-hospital-admissions-The-COVID-Symptom-S/tree/main/proj/sens2020559/COVID-19/Codes> (CSSS_codes.zip). The code describes all cleaning and pre-processing steps that were applied to the data in this study:

1. Variable names were harmonized across the three raw data files from the COVID Symptom Study app in both COVID Symptom Study Sweden (CSSS) and ZOE COVID Study: a) patient data (baseline information on study participants), b) symptom assessments data, and c) COVID-19 test results data.
2. In CSSS, data was limited to recordings between April 29, 2020 to February 10, 2021.
3. In ZOE COVID Study, data was limited to recordings between March 30, 2020 to February 10, 2021.
4. Participants in both CSSS and ZOE COVID Study were excluded if they: 1) never submitted a daily report, 2) had missing age or reported an age <18 years or >99 years, or 3) stated their sex as other or intersex.
5. Participants in both CSSS and ZOE COVID Study whose last report was within seven days of joining the study or did not provide a valid postal code were excluded from the prevalence estimation. However, they were included in model training if they reported a PCR test and submitted at least one symptomatic daily report within seven days preceding or on the test date.
6. Participants in both CSSS and ZOE COVID Study with long-lasting COVID-19 symptoms were excluded after their 30th day of reporting loss of smell and/or taste to ensure that the estimates were not inflated due to post-acute sequelae of COVID-19.
7. In both CSSS and ZOE COVID Study, symptoms reported during the first seven days after a participant had joined the study were excluded.
8. In both CSSS and ZOE COVID Study, in case of gaps in daily reporting, the last report was assumed to be current for no more than seven days.
9. To derive county information for all study participants in CSSS, we matched their stated postal codes with a file from the external company "Postnummerservice" that translated postal codes into county codes.
10. To derive county information for all study participants in ZOE COVID Study, a file connecting postal codes with UK NHS health regions (as of February 2021) was obtained from the Office for National Statistics.

11. The name of the variable (“rgn”) was changed to “county” to better harmonize with the code for the Swedish data.
12. A smaller number of postal codes areas in Sweden as well as in England overlapped regions. In the Swedish dataset, postal codes areas were assigned to the county to which they overlapped most, by webscraping from the homepage <https://postnummer.cybo.com/sverige/#listcodes>. We then converted the county codes (in the variable “länskod”) presented as digits to county names using a key file from Statistics Sweden. In the English dataset, overlapping areas were randomly assigned to one of the two NHS health regions.
13. The “clock” variable was converted from string to date format.
14. Symptom assessments were converted from string format to numeric values (1 if the symptom was present, and 0 if the symptom was absent at each update).
15. Multiple symptom assessment entries within the same day were collapsed, yielding a single row per participant and day with the value “1” if the symptom was marked as “1” at least once and else “0”
16. In the COVID test results data, free text containing specific keywords were reclassified as binary variables for PCR and antibody tests (1=present, 0=absent). The PCR keywords were: “swab”, “pinne”, “svalg”, “svalj”, “tops”, “spit_tube”, “spott”, “Naso”, “hals”, “saliv”, “PCR”, “sputum”, “Nose”, “näs”, “Nadophar”, “throat”, “Nasalt/Oralt” and “region Skånes testkit”. The antibody keywords were “blod”, “blood”, “antikr”, “stick”, “kapill”, “stack i”, “immun”, “IgG”, “Imuni”, “Finger”, “IgM”, “Venprov”, “armv”, “anitkropp”, “kapill”, “Kaplär”, “Serologiskt” and “venös”.
17. The variable names in ZOE COVID Study were modified to match the variable names in CSSS to facilitate analysis using the same code.
18. Age was calculated by subtracting the year of birth provided by the participant from the year the participant created their profile. Individuals with implausible ages were removed when filtering out ages outside the age range of the study (18-99).
19. BMI was calculated from the pre-existing variables for weight and height as weight (kg)/(height (m)²). We labelled self-reported height, weight, and body mass index (BMI) as “missing” if <130 or >210 cm, <35 or >300 kg, and <15 or >70kg/m², respectively
20. All variables present in CSSS and ZOE COVID Study not included in the current analyses were excluded from the analysis data set to enable shorter analyses time.
21. The large ZOE COVID Study dataset was subdivided into 16 parts (2-week intervals) to facilitate analysis, with the last observation carried forward for those that did not report in

the start of the part of the dataset. The maximum number of days an observation was carried forward was set to 7.

22. Once all three files (patient data, symptom assessments data, COVID-19 test results data) from CSSS and ZOE COVID Study, respectively, were harmonized with matching variable names, they were merged together into one file for each country.

Updated paragraph in the Methods section of the manuscript, pages 17-18:

“Study population and data management

In this study, we included data from April 29, 2020 to February 10, 2021. Participants were in this study excluded from the analyses if they: 1) never submitted a daily report (n=5,931), 2) had missing age or reported an age <18 years or >99 years (n=801), or 3) stated their sex as other or intersex (n=236) as this sample size was insufficient for a further analysis (Supplementary Table 8). Participants whose last report was within seven days of joining the study (n=45,483) or did not provide a valid postal code (n=7,310) were excluded from the prevalence estimation, but included in model training if they reported a PCR test and submitted at least one symptomatic daily report within seven days preceding or on the test date (n=967). The final study population consisted of 143,531 individuals (Figure 6). We labelled self-reported height, weight, and body mass index (BMI) as “missing” if <130 or >210 cm, <35 or >300 kg, and <15 or >70kg/m², respectively. “Obesity” was defined as BMI ≥30kg/m². Additional detailed information on data cleaning and pre-processing is available in the Supplementary Material.”

New section in the Supplementary Material, pages 5-6:

“Data cleaning and pre-processing

The pre-processing code “2_preprocess_abbrev.R” is available at [https://github.com/ulfha881/App-based-COVID-19-syndromic-surveillance-and-prediction-of-hospital-admissions-The-COVID-Symptom-S/tree/main/proj/sens2020559/COVID-19/Codes\(CSSS_codes.zip\)](https://github.com/ulfha881/App-based-COVID-19-syndromic-surveillance-and-prediction-of-hospital-admissions-The-COVID-Symptom-S/tree/main/proj/sens2020559/COVID-19/Codes(CSSS_codes.zip)). The code describes all pre-processing steps that were applied to the data in this study:

1. Variable names were harmonized across the three raw data files from the COVID Symptom Study app in both COVID Symptom Study Sweden (CSSS) and ZOE COVID Study: a)

patient data (baseline information on study participants), b) symptom assessments data, and c) COVID-19 test results data.

2. In CSSS, data was limited to recordings between April 29, 2020 to February 10, 2021.

3. In ZOE COVID Study, data was limited to recordings between March 30, 2020 to February 10, 2021.

4. Participants in both CSSS and ZOE COVID Study were excluded if they: 1) never submitted a daily report, 2) had missing age or reported an age <18 years or >99 years, or 3) stated their sex as other or intersex.

5. Participants in both CSSS and ZOE COVID Study whose last report was within seven days of joining the study or did not provide a valid postal code were excluded from the prevalence estimation. However, they were included in model training if they reported a PCR test and submitted at least one symptomatic daily report within seven days preceding or on the test date.

6. Participants in both CSSS and ZOE COVID Study with long-lasting COVID-19 symptoms were excluded after their 30th day of reporting loss of smell and/or taste to ensure that the estimates were not inflated due to post-acute sequelae of COVID-19.

7. In both CSSS and ZOE COVID Study, symptoms reported during the first seven days after a participant had joined the study were excluded.

8. In both CSSS and ZOE COVID Study, in case of gaps in daily reporting, the last report was assumed to be current for no more than seven days.

9. To derive county information for all study participants in CSSS, we matched their stated postal codes with a file from the external company "Postnummerservice" that translated postal codes into county codes.

10. To derive county information for all study participants in ZOE COVID Study, a file connecting postal codes with UK NHS health regions (as of February 2021) was obtained from the Office for National Statistics.

11. The name of the variable ("rgn") was changed to "county" to better harmonize with the code for the Swedish data.

12. A smaller number of postal codes areas in Sweden as well as in England overlapped regions. In the Swedish dataset, postal codes areas were assigned to the county to which they

overlapped most, by webscraping from the homepage

<https://postnummer.cybo.com/sverige/#listcodes>. We then converted the county codes (in the variable "länskod") presented as digits to county names using a key file from Statistics Sweden. In the English dataset, overlapping areas were randomly assigned to one of the two NHS health regions.

13. The "clock" variable was converted from string to date format.

14. Symptom assessments were converted from string format to numeric values (1 if the symptom was present, and 0 if the symptom was absent at each update).

15. Multiple symptom assessment entries within the same day were collapsed, yielding a single row per participant and day with the value "1" if the symptom was marked as "1" at least once and else "0"

16. In the COVID test results data, free text containing specific keywords were reclassified as binary variables for PCR and antibody tests (1=present, 0=absent). The PCR keywords were: "swab", "pinne", "svalg", "svalj", "tops", "spit_tube", "spott", "Naso", "hals", "saliv", "PCR", "sputum", "Nose", "näs", "Nadophar", "throat", "Nasalt/Oralt" and "region Skånes testkit". The antibody keywords were "blod", "blood", "antikr", "stick", "kapill", "stack i", "immun", "IgG", "Imuni", "Finger", "IgM", "Venprov", "armv", "anitkropp", "kapill", "Kaplär", "Serologiskt" and "venös".

17. The variable names in ZOE COVID Study were modified to match the variable names in CSSS to facilitate analysis using the same code.

18. Age was calculated by subtracting the year of birth provided by the participant from the year the participant created their profile. Individuals with implausible ages were removed when filtering out ages outside the age range of the study (18-99).

19. BMI was calculated from the pre-existing variables for weight and height as $\text{weight (kg)} / (\text{height (m)}^2)$. We labelled self-reported height, weight, and body mass index (BMI) as "missing" if <130 or >210 cm, <35 or >300 kg, and <15 or >70kg/m², respectively

20. All variables present in CSSS and ZOE COVID Study not included in the current analyses were excluded from the analysis data set to enable shorter analyses time.

21. The large ZOE COVID Study dataset was subdivided into 16 parts (2-week intervals) to facilitate analysis, with the last observation carried forward for those that did not report in the

start of the part of the dataset. The maximum number of days an observation was carried forward was set to 7.

22. Once all three files (patient data, symptom assessments data, COVID-19 test results data) from CSSS and ZOE COVID Study, respectively, were harmonized with matching variable names, they were merged together into one file for each country.”

Comment 4: A completed TRIPOD checklist wasn't found in the submission. This should be provided.

Authors' reply: Thank you. A completed TRIPOD checklist for Prediction Model Development and Validation has now been included in our submission.

Reviewer #2

I am fine with the revised manuscript.

Authors' reply: Thank you.

Reviewer #3

I commend the authors for the very thorough revision. The revised manuscript addresses the comments appropriately, and provides additional explanation, details and transparency regarding methods and results.

Authors' reply: We would like to thank the Reviewer for these kind and encouraging remarks.

Reference

1. DeLong ER, DeLong DM, Clarke-Pearson DL. Comparing the Areas under Two or More Correlated Receiver Operating Characteristic Curves: A Nonparametric Approach. *Biometrics* **44**, 837-845 (1988).